complexity/environmental science

urban facilities, Taylor's Law, spatial aggregation, factor decomposition

**Author for correspondence:**
Xin Yan
e-mail: yanxinbob@qq.com

# Taylor's power law and its decomposition in urban facilities

## Liang Wu, Chi Gong and Xin Yan

The School of Economics, Sichuan University, 29 Wangjiang Road, Chengdu 610065, People's Republic of China

 LW, 0000-0002-3606-356X; XY, 0000-0002-6818-2635

As one of the few generalities in ecology, Taylor's power law admits a power function relationship $V = aM^b$ between the variance $V$ and mean number $M$ of organisms in a quadrat. We examine the spatial distribution data of seven urban service facilities in 37 major cities in China, and find that Taylor's Law is validated among all types of facilities. Moreover, Taylor's Law is robust if we shift the observation window or vary the size of the quadrats. The exponent $b$ increases linearly with the logarithm of the quadrat size, i.e. $b(s) = b_0 + A \log(s)$. Furthermore, the ANOVA test indicates that $b$ takes distinct values for different facilities in different cities. We decompose $b$ into two different factors, a city-specific factor and a facility-specific factor (FSF). Variations in $b$ can be explained to a large extent by the differences between cities and types of facilities. Facilities are more evenly distributed in larger and more developed cities. Competitive interchangeable facilities (e.g. pharmacy), with larger FSFs and smaller $b$s, are less aggregated than complementary services (e.g. restaurants).

## 1. Introduction

Bettencourt & West [1] state: '... cities are remarkably robust: success, once achieved, is sustained for several decades or longer, thereby setting a city on a long run of creativity and prosperity' (p. 913). The urban facilities such as convenience stores, restaurants, schools etc., play critical roles to set a city on a long run of creativity and prosperity by providing services to meet citizens' basic needs. The spatial distribution of facilities demonstrates rather different patterns on the map, e.g. restaurants aggregate into clusters, while pharmacies are dispersed across the whole city. The location of facilities can be easily affected by city geographical landscapes, business nature, local consumer characteristics, etc. It is a complex task to understand the commercial logics of the locations of the urban facilities. Some statistical regularities can be found if we take

spatial statistical approaches on the location data by ignoring other details. Pablo [2] uses a network approach to reveal many important facts about the commercial organization of retail trade based on location data alone.

Urban facilities resemble organisms of a species in an area, in the sense that the same type of facilities in a city may help each other survive, while at the same time, they compete for various resources. As a fundamental law in ecology, Taylor's Law characterizes the fluctuation of the organisms' spatial distribution. Based on the spatial data of urban facilities through the API of Baidu[1] digital map, we are able to test whether Taylor's Law can be applicable to urban facilities, and then test if the exponents of Taylor's Law can be used to quantify the characteristics of the spatial distribution of different facilities by removing the city factor from the total exponent embedded with both city and facility factors.

Taylor [3] describes a power function relationship $V = aM^b$ between the between-sample variance in density $V$ and the overall mean density $M$ of a sample of organisms in an area. The exponent $b$ describes the heterogeneity of the spatial or temporal distribution. For example, $b = 1$ corresponds to a Poisson distribution, $b > 1$ indicates the clumping of organisms.

Besides the distribution of organisms, Taylor's power law has found application to seemingly unrelated phenomena like human sexual pairing [4], human haematogenous cancer metastases, the clustering of childhood leukaemia [5], measles epidemics [6] and gene structures [7]. Taylor's Law has recently been increasingly applied in social systems modelling, e.g. in [8] Taylor's Law is used for the analysis of human spatial behaviour. Given such broad applicability of Taylor's Law in many seemly mysterious and complicated natural processes, one might ask whether there is any general principle at the basis of all these processes.[2] A large body of literature has been devoted to this question, and many theoretical models have been introduced to explain Taylor's Law. For instance, Andersen *et al.* [9] propose a Markovian population model and justify this model through simulations, which show that with the increase in the average population density, the variance to mean relationship would approach a power function with a maximum exponent value 2. Fronczak & Fronczak [10] suggest that Taylor's Law is a result from the second law of thermodynamics and the behaviour of the density of states. Kendal & Jørgensen [11] further notice that the cumulant generating function derived by Fronczak & Fronczak [10] is known as the Tweedie distribution and therefore suggests that Taylor's power law results from the central-limit-theorem-like convergence.

Despite those theoretical attempts to explain the origin of the power law, however, it has not come to agreement on the meaning of the exponent $b$. Empirical evidence is clearly needed if we can observe how $b$ varies among different species at the same location, and similarly how $b$ varies for the same species across different locations. In this aspect, our study might shed light on the meaning of $b$ by exploring the potential mean–variance relationship of Taylor's Law in the spatial distribution of urban facilities. We will measure $b$ for various urban facilities in many cities. It is understood that the measurement of scaling laws is subject to inaccurate estimations [12,13]. The advantage of our approach is that we can measure $b$ for many cities and various urban facilities. One measurement of $b$ for one urban facility in a city may be uncertain. The combined statistical test, which uses many $b$s for different combinations of facilities and cities, can yield reasonably robust results.

Our statistical test shows the values of $b$ are different from one city to the other city; they are also different from one facility to the other facility. Both a city-specific factor (CSF) and a facility-specific factor (FSF) are embedded in $b$. In order to study these two key factors contributing to the difference among $b$s and explore the mechanism underlying the distribution of urban facilities, we decompose the inverse of exponent $b$ by examining their contribution to the numbers of facilities located in a study region. The CSF plus the FSF accounts for a remarkably high level of over 92% of $1/b$, even though there are inevitably estimation errors in $b$. The meaning of CSF and FSF are also discussed in the paper. The CSF reflects the overall density of all the facilities in a city, as well as their aggregation level of all types of facilities put together. FSF indicates the overall aggregation level of each type of facility in all the cities. For example, the first tier four cities rank in the top four in the CSF values due to the high overall facility density in these cities, while restaurant tops the overall aggregation level due to their complementary nature. These findings are consistent with our intuitive understandings of these cities and urban facilities.

---

[1]Baidu is the largest search engine and digital map service provider like Google in China.

[2]For more detailed and in-depth review of relevant literature about Taylor's Law and the explanation offered, refer to [7].

As suggested by Leitão *et al.* [14], city-specific observations scale nonlinearly with population, we also check the CSF for cities of different population size, and find that larger and more developed cities tend to have smaller $b$s. It means that in larger and more developed cities, we do not only have more service facilities available to the citizen, but also the facilities are more evenly distributed to provide more convenient services to citizens.

Serving as areas for the concentration of human activities, cities are considered to be the principal engines of innovation and economic growth [15,16]. Today, more than half of the world population live in cities. The developed world is now about 80% urbanization and the entire planet will follow this pattern by around 2050, with some two billion people moving to cities, especially in China, India, Africa and Southeast Asia.[3] Countries around the world are experiencing a rapid urbanization process, which presents an urgent challenge for developing predictive, subtle and quantitative theories and methods, providing necessary technical support for urban organization and sustainable development.

As consumers of energy and resources and producers of artefacts, information and waste, cities have often been compared with biological entities and ecosystems [17–19]. Bettencourt *et al.* [17] show that there are very general and non-trivial quantitative regularities of social activities common to all cities across urban systems, and many diverse properties of cities, such as patent production, personal income and crime, are shown to be power law functions of population size. Besides, the size distribution of cities fits a power function (known as Zipf's Law): the number of cities with populations greater than $S$ is proportional to $1/S$ [20]. Geometrically, the complex spatial structure of cities have apparent fractal nature associated with individual cities and entire urban systems [21]. Through exploring possible consequences of the scaling relations by deriving growth equations, Bettencourt & West [1] quantify the dramatic difference between growth fuelled by innovation versus that driven by economies of scale, suggesting that as population grows, major innovation cycles must be generated at a continually accelerating rate, so as to sustain growth and avoid stagnation or collapse. Bettencourt & West [1] state that cities should be treated as a complex dynamic system, which is capable of aggregating and manifesting human cognitive ability, leading to open socioeconomic development.

The contribution of this paper mainly lies in the following aspects. This is the first paper revealing Taylor's Law in the spatial distribution of urban facilities. Furthermore, we discuss the size effect of quadrats and discover that the exponent $b$ increases linearly with the logarithm of the size of the quadrat, i.e. $b(s) = b_0 + A \log(s)$, which has not been documented in the previous literature. Moreover, we decompose the inverse of exponent $b$ to examine two different factors contributing to the numbers of facilities in a study region (we call it a quadrat) within a city, and find that both the CSF and the FSF have their own concrete and specific implications.

This paper proceeds as follows. Section 2 introduces the source of the data and data processing method. Sections 3 and 4 present the empirical results of Taylor's Law and size effects of quadrats. Section 5 discusses the size effect of quadrats. Section 6 decomposes the factors affecting the number of a facility in the quadrats within a city. Section 7 discusses the results and concludes.

## 2. Data preparation

Baidu provides a programmable interface to use the digital map. We collect the spatial coordinates data by calling the interface for seven types of service facilities, in the city area and adjacent counties of 37 major cities in China. The seven types of facilities are: beauty salons, banks, stadiums, schools, pharmacies, convenience stores and restaurants.[4] The exact meaning of these facilities is described in table 1. The 37 major cities consist of four direct-controlled municipalities (Beijing, Shanghai, Chongqing and Tianjin), 30 provincial capitals and sub-provincial cities and three other large cities with a high ranking in GDP output. These cities are the largest Chinese cities in both population and aggregate economical output. Population size and number of facilities which are studied in this paper are listed in table 7.

[3]UN-Habitat. State of the World's Cities 2010/2011—Cities for All: Bridging the Urban Divide (2010); available at http://www.unhabitat.org.

[4]We focus our study on services, because they tend to be located in dense areas [22]. Besides, we only consider the private sector, due to its evolving competitive nature driven by the market. Therefore, we exclude hospitals as many health facilities are state owned. The considered facilities in this study are representative in the sense that they are related to our daily lives in different aspects.

**Table 1.** Descriptions of urban facilities.

| facilities | descriptions |
| --- | --- |
| beauty salons | shops provide service for hairdressing, manicuring or other cosmetic treatment for men and women |
| banks | branches of banks and self-service banking centres |
| stadiums | large comprehensive sports arena and specialized sports facility such as swimming pool |
| schools | kindergarten, primary school, middle school, college and other educational facilities |
| pharmacies | drugstore, a place where medicines are compounded or dispensed |
| convenience stores | small shops offering household products |
| restaurants | places serving meals, all types included |

The raw data include the latitude and longitude coordinates of each facility. For illustrative purposes, we randomly choose 300 samples for each of the seven types of facilities in Beijing and mark these samples in figure 1. We then convert the raw data of the latitude and longitude coordinates into planar coordinates in the unit of metre, so as to facilitate the calculation of distance and area selection. It is hard to define the exact boundary of a city. We select a central point $(lng_0, lat_0)$ by examining the satellite map of a city, and use the central point as our origin of the planar coordinate system.[5] We then convert the spherical coordinate of each facility to the corresponding planar coordinate. For example, given the latitude and longitude coordinate $(lng_i, lat_i)$ of a facility, its planar coordinates $(x_i, y_i)$ are calculated by

$$\left. \begin{array}{l} x_i = \text{Distance}((lng_i, lat_0), (lng_0, lat_0)) \times \text{Sign}(lng_i - lng_0) \\ y_i = \text{Distance}((lng_0, lat_i), (lng_0, lat_0)) \times \text{Sign}(lat_i - lat_0), \end{array} \right\} \qquad (2.1)$$

where Sign is a sign function, with $\text{Sign}(x) = 1$ if $x > 0$, $\text{Sign}(x) = 0$ if $x = 0$, and $\text{Sign}(x) = -1$ if $x < 0$. The Distance function is defined as the length of the largest arc connecting two points on the spherical surface of the earth. The radius of the earth is used as $R = 6\,371\,004$ m.

It is hard to draw a boundary to separate a city from its suburbs since the city–suburbs lines are rather blurring in most cities. In this study, our strategy is to combine an initial choice and a later examination of the data. The initial choice is to set a large enough starting area to cover most populated city zones and to prevent the boundary from being drawn into adjacent towns. Then in the starting area, we draw a grid of non-overlapping sub-areas. A later examination is to detect if a sub-area should be discarded if the number of facilities in that area is less than a threshold. The size of the starting area is set as $40\,000 \times 40\,000$ m,[6] as a stating point centred at the city central point.[7] This area is then divided into sixteen $10\,000 \times 10\,000$ m non-overlapping sub-areas.[8] However, because of the irregularity of the city area, a few of the sub-areas could be corresponding with the depopulated zones. We mark a sub-area as not valid if the total number of a given facility is less than 20.[9] This method naturally decides where we should draw a line for the city boundary according to the concentration of facilities.

[5]For instance, the latitude and longitude coordinate for the central point of Beijing is (116.413648 E, 39.913561 N).

[6]The choice of $40\,000 \times 40\,000$ m for the starting area is because it is large enough to cover most populated area of a city. The choice of this number should not be an issue for most cities. If we make it bigger, newly added area will most likely be identified as non-city zone. For some cities, it is possible that we can get a bigger exponent $b$ if we enlarge the area since we might include less populated adjacent towns.

[7]The central point is marked by hand for each city according to the satellite image.

[8]The choice of 16 is to balance the need to get more data points and less estimation error for the mean–variance pair. If the number is too big, say, 25, we may end up with small number of facilities in each sub-area, which leads to large estimation error of the mean and variance. If the number is too small, say, 9, for some irregular cities, we may not have enough data points to draw a line for the mean and variance.

[9]The choice of the threshold number 20 to decide whether a sub-area is valid does not affect the validity of Taylor's Law. However, if the number is set too small, then in some sub-areas, the mean and variance are inaccurate since there are not many facilities in each quadrat. On the other hand, if the number is set too big, we may not have enough cities which have at least five valid sub-areas so that we can estimate the values for the exponent $b$ for all considered seven types of facilities. In our case, the threshold number is set as 20 to balance the above two issues based on the data we have, we end up having 23 cities with all seven $b$s estimated.

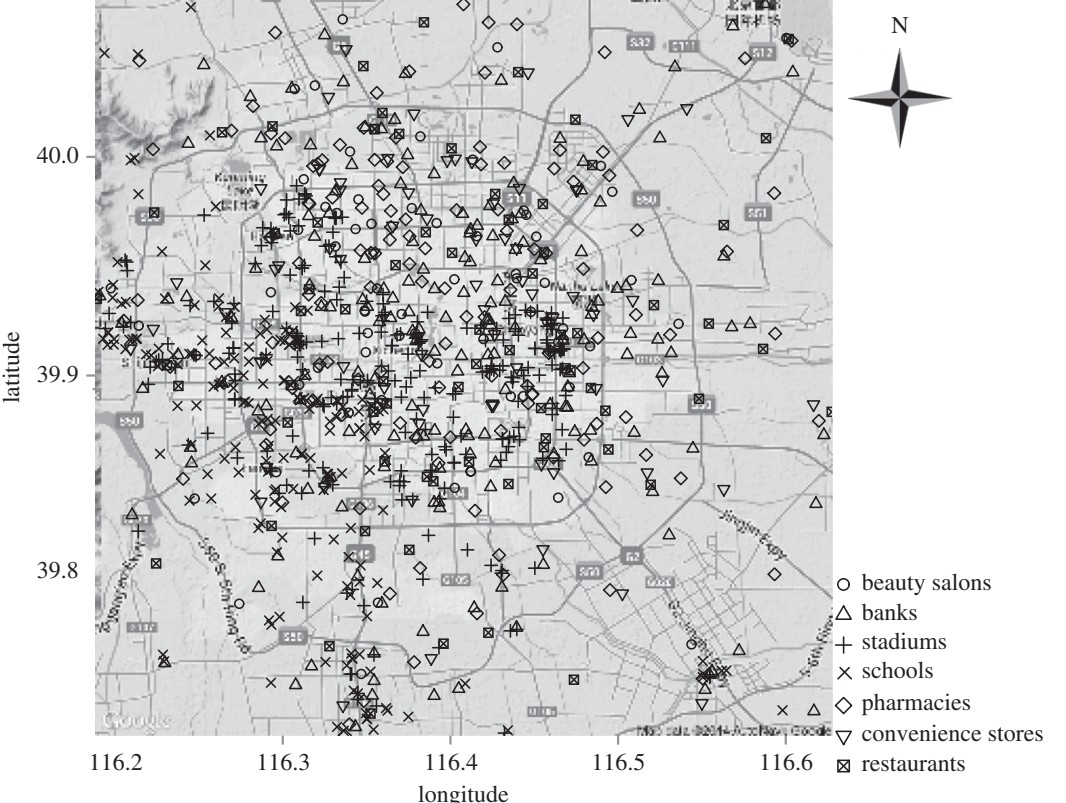

**Figure 1.** The distribution of seven types of facilities in Beijing. Three hundred random samples are marked in the map for illustrative purposes.

Each sub-area is further divided into $J \times J$ small quadrats to get an estimation of mean and variance.[10] For example, if $J = 5$, a quadrat is a $2000 \times 2000$ m area. We denote the number of a facility in quadrat $j$ of sub-area $i$ with $N_{ij}$, where $i = 1, 2, \ldots, 16$ and $j = 1, 2, \ldots, J \times J$. For each of the 16 sub-areas, we calculate the mean and variance of the number of a facility, thus getting a maximum of 16 pairs of mean and variance denoted by $(M_i, V_i)$, where $M_i = E[N_i]$ and $V_i = \mathrm{Var}(N_i)$ $(i = 1, 2, \ldots, 16)$ are computed using values of $N_{i,j}$, $j = 1, 2, \ldots$ for each sub-area $i$. In order to get a good estimation of mean and variance in each sub-area $i$, $N_{i,j}$ should be bigger than 0 for a sufficient number of quadrats. In some cities, the numbers of certain facilities are not large enough, which may greatly increase the size of error in the estimation of the means and variances. To avoid this problem, we decide that if the total number of valid sub-areas is small, e.g. less than five (i.e. we have less than five pairs of means and variances), we cannot get a reasonably good estimation of the mean–variance relationship. We then have to give up our estimation for this given combination of facility and city.

## 3. Taylor's power law

It should be noted that in our study, every pair of mean and variance is estimated in an area using the same quadrat size. Hence, the potential relationship between the means and variances only reflects the fluctuation of the number of events in a quadrat of the given size, not a scaling rule where the number of events is measured in a series of expanding quadrats discussed by Wu *et al.* [23]. Through examining whether or not there is linear correlation between the natural logarithm of the means and variances, i.e. $\log V = \log a + b \log M$, we can judge whether or not Taylor's power law is applicable to the urban facilities.

Taking Beijing as an example, we draw the scatter plots of the means and variances for beauty salons, stadiums, schools and banks in figure 2a for $J = 5$ and figure 2b for $J = 8$. Remember that a sub-area is divided into $J \times J$ quadrats. $J = 5$ means that each quadrat in which the number of facilities $N_{i,j}$ is

[10]We discuss how the choice of $J$ affect our results in §4.

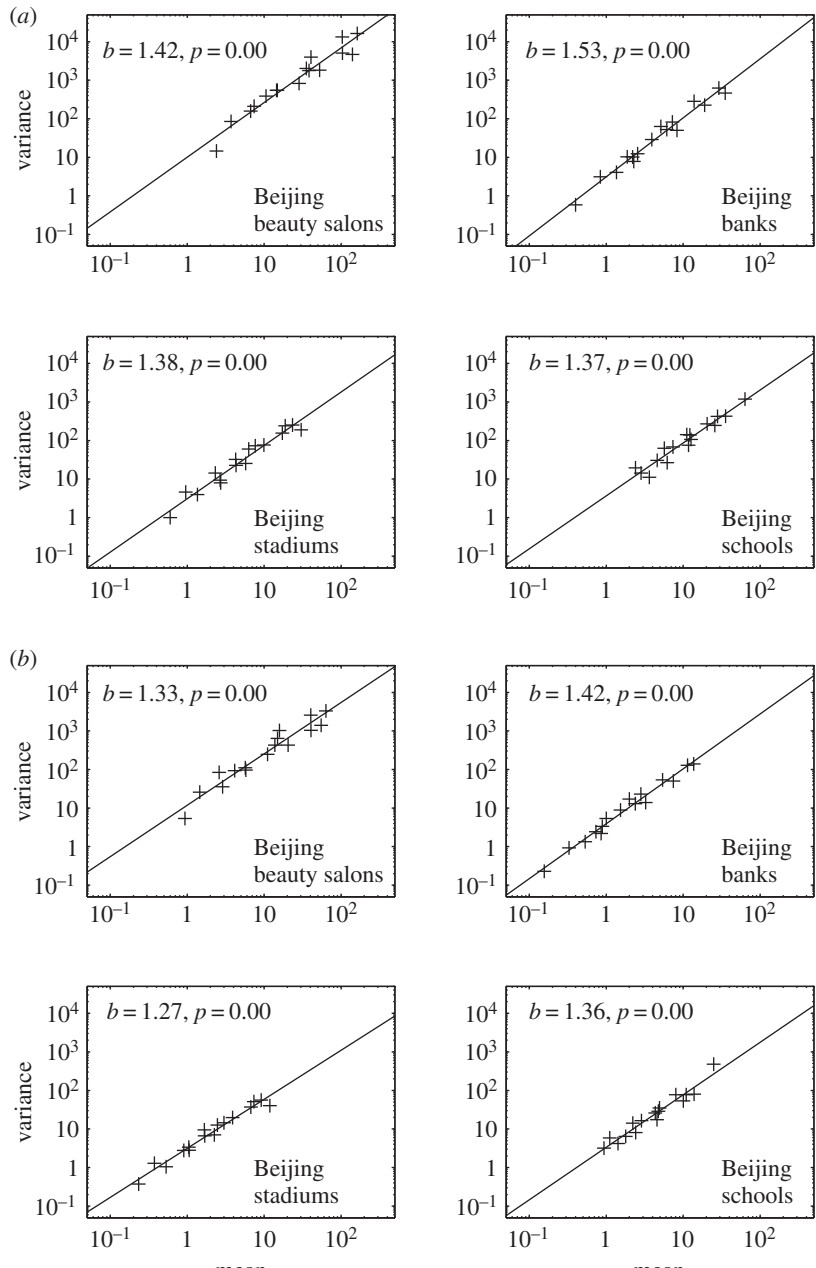

**Figure 2.** Scatter plots for beauty salons, banks, stadiums and schools in Beijing when the size of the quadrats is (*a*) 2000 × 2000 m (*J* = 5) and (*b*) 1250 × 1250 m (*J* = 8). The fitted exponent *b* and its *p*-value are included.

counted is 2000 × 2000 m. $J = 8$ means that the size of each quadrat is 1250 × 1250 m. As we can see from figure 2, for each facility, the means and variances are distributed along a straight line in the log–log plot, which clearly indicates that it satisfies Taylor's Law.

The facilities, for which we can have more than five pairs of mean and variance to get a reasonable estimation of Taylor's Law, are different from one city to the other city. There are 23 cities in which we can estimate Taylor's Law for all the seven types of facilities. In order to give a clear picture of the exponent *b* for each facility, we put together the values of exponent *b* for each facility in those 23 cities in figure 3. Here, $J = 5$ is used. We can see from figure 3 that almost all the values of *b* fall between 1 and 2, with a concentration around 1.5. $b > 1$ indicates that the facilities are aggregated. There are two cases for pharmacies that $b < 1$, it means that pharmacies in those two cities tend to be more uniformly distributed due to their competitive nature. There is a large fluctuation in the values of exponent *b* for different combination of facilities and cities. The main reason for the fluctuation lies in the fact that the distribution of urban facilities are driven by business nature of

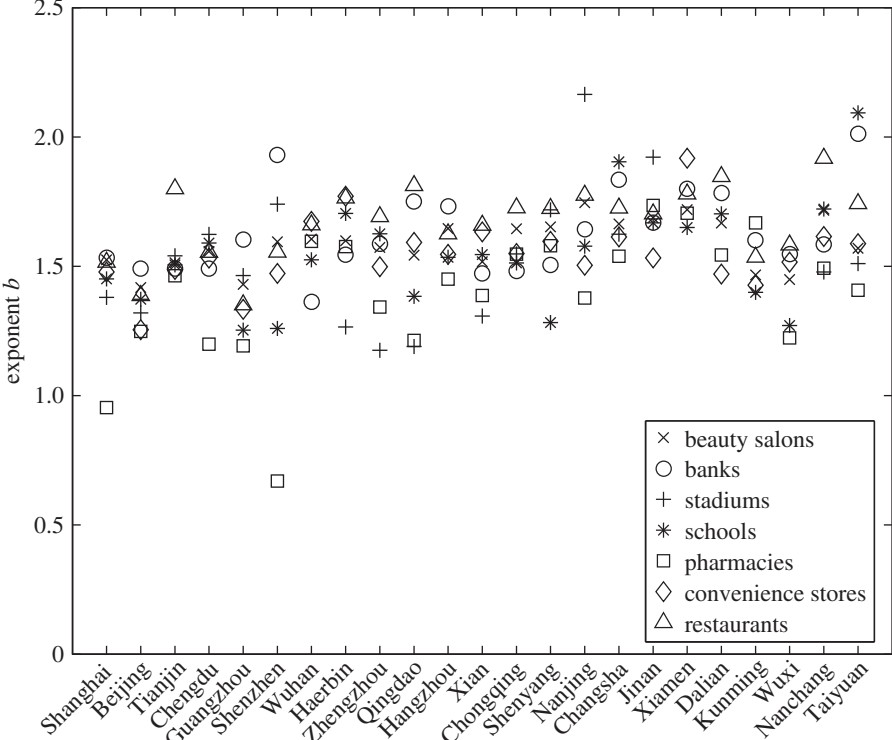

**Figure 3.** The values of exponent $b$ for 23 cities when $J = 5$. The fluctuation of the exponent mainly lies in the distinct business nature of facilities and different social-economical environment of cities. Over 92% of the fluctuation can be explained after we decompose the exponent into a CSF and an FSF.

**Table 2.** Exponent of Taylor's Law for different choices of $J$ and when the quadrats are moved up and to the right by half of the quadrat size. Here $b$ and $b_s$ refers to the exponent of Taylor's Law before and after the quadrats are shifted. The $p$-value corresponds to the $T$-test for the null hypothesis $b = b_s$.

| $J$ | 4 | 5 | 6 | 7 | 8 |
| --- | --- | --- | --- | --- | --- |
| mean (s.d.) of $b$ | 1.62 (0.22) | 1.56 (0.20) | 1.51 (0.21) | 1.46 (0.21) | 1.42 (0.21) |
| mean (s.d.) of $b_s$ | 1.59 (0.19) | 1.53 (0.21) | 1.48 (0.20) | 1.45 (0.18) | 1.41 (0.19) |
| $p$-value | 0.16 | 0.18 | 0.22 | 0.59 | 0.62 |

urban facilities, as well as social-economical environment of a city. We will decompose the exponent in the later part of the paper, which can explain up to 92% of the fluctuations. The rest may be attributed to the estimation error.

We organize the cities along the horizontal axis by population size from the highest to the lowest. One should understand that the population data are not accurate due to the rapid urbanization process in China. The population data reported in this study only include permanent residents, it does not include the migrating population who do not have the city Hukou. There is a slight tendency of increase in exponent $b$ from the left to the right. It seems that bigger and more developed cities have smaller values of $b$. In the next section, after we decompose the exponent $b$, we will plot the CSF against the population size.

As a robust check, we will test if Taylor's Law still holds and how the exponent $b$ varies if we shift the locations of the observation windows of the quadrats.

The whole study area of a city is covered by non-overlapping quadrats. For different choice of $J$ (we choose $J = 4, \ldots, 8$), the size of the quadrat is $L = 10\,000/J$. We notice that for different choice of $J$, Taylor's Law still holds by observing that the variance is a power function of the mean. Figure 2 gives an illustration of the power law relationship for $J = 5$ and $J = 8$. The results are similar for other choices of $J$.

Then we shift the location of the observation window to the maximum by moving the window up by $L/2$ then to the right by $L/2$. By using the seven types of facilities in the 23 cities, we report the mean and standard deviation of $b$ before we move the window and after we move the window in table 2. We then take $T$-test to test the null hypothesis that $b$ does not change if we shift the observation window. The $p$-values are larger than 0.05 for all $J$s, indicating that there is no significant difference in $b$. We will discuss how $b$ varies when we use different sizes of the quadrats in the next subsection.

# 4. How does $b$ vary with size of the quadrat

Note that in this paper, the mean–variance relationship is obtained when the density of the events fluctuates across different sub-areas of a city, while the size of the quadrat is kept fixed. For different choices of $J$ (hence different sizes of the quadrats), we have demonstrated in the previous section that Taylor's Law still holds. We have the following proposition on how the exponent $b$ varies with the size of the quadrat. It is a result that can be derived from the combination of two scaling laws regarding mean–variance relationship, namely, Taylor's Law and size-scaling law as studied in [23].

**Proposition 4.1** *The exponent $b$ of Taylor's Law measured in quadrats of size $s$, we have $b(s) = b_0 + A \log(s)$, where $b_0 \equiv b(s = 1)$ is the exponent measured in quadrats of unit size, and $A$ is a constant to be decided by experiment.*

It should be noted that $s$ is dimensionless, which does not have a unit. In other words, $s$ is relative, measuring the ratio of the size of a quadrat relative to the base quadrat for which we let $s = 1$. For instance, at two different scales, we pick one and let the size be $s = 1$, the size of the other one relative to the first one is $s$. The formula tells how the exponents at the two different scales are related.

To prove the proposition, we need to cite the other scaling invariant mean–variance relationship as studied in [23], if we vary the size of the quadrats while the density (i.e. number of events per unit area) is fixed. If we scale the size of the quadrats by $s$, the mean in the new quadrats is scaled by $M(s) \propto s^2$ in a two-dimensional space. Owing to the power law of the covariance density, i.e. the covariance density $\gamma(r)$ is a power function of the physical distance $r$ between two locations, $\gamma(r) \propto r^{-c}$. It is shown in [23] that the variance is scaled by $V \propto s^{4-c}$. Thus, the mean–variance relationship is given by a power law $V \propto M^{b'}$ with $b' = 2 - c/2$. To differentiate the two power laws, we refer to the latter one as size-scaling law, and Taylor's Law as the density-scaling law. We use $b'$ and $b$ to refers to the exponent of the size-scaling law and the density-scaling law, respectively.

Assume that there are two sub-areas $i$ and $j$ with different densities. The density (i.e. the number of events in a unit area) in the two sub-areas is $M_i$ and $M_j$, respectively. First we use quadrats of unit size to measure the mean and variance. According to Taylor's Law, we have

$$\log(V_i) - \log(V_j) = b_0(\log(M_i) - \log(M_j)). \tag{4.1}$$

Then, we increase the quadrat size by $s$. In the new quadrats of sub-area $i$ and $j$, the mean is $M_i(s) = s^2 M_i$ and $M_j(s) = s^2 M_j$, respectively. According to the size-scaling law as shown in [23], we have the following relationship, respectively, for sub-area $i$ and $j$,

$$\log(V_i(s)) - \log(V_i) = b_i'(\log(s^2 M_i) - \log(M_i)) \tag{4.2}$$

and

$$\log(V_j(s)) - \log(V_j) = b_j'(\log(s^2 M_j) - \log(M_j)), \tag{4.3}$$

where $V_i(s)$ and $V_j(s)$ are the variance using the new quadrats (with size $s$) in sub-area $i$ and $j$, respectively. Note that $V_i$ and $V_j$ are measured in the base quadrats when $s = 1$. $b'$ refers to the exponent of the size-scaling law. It may take different values in different sub-areas with different densities, so we use subscript $i$ and $j$ to refer to the difference.

Combining the above three equations, we have

$$\log(V_i(s)) - \log(V_j(s)) = \log(s^2)(b_i' - b_j') + b_0(\log(M_i(s)) - \log(M_j(s))). \tag{4.4}$$

Since Taylor's Law still holds when the quadrats are increased in size by $s$, we have the following two possible cases.

Case 1, $b_i' - b_j' = 0$. The size-scaling law of mean–variance does not depend on the density of the events. Then $\log(V_i(s)) - \log(V_j(s)) = b_0(\log(s^2 M_i) - \log(s^2 M_j))$. Thus $b(s) = b_0$, i.e. we have the same

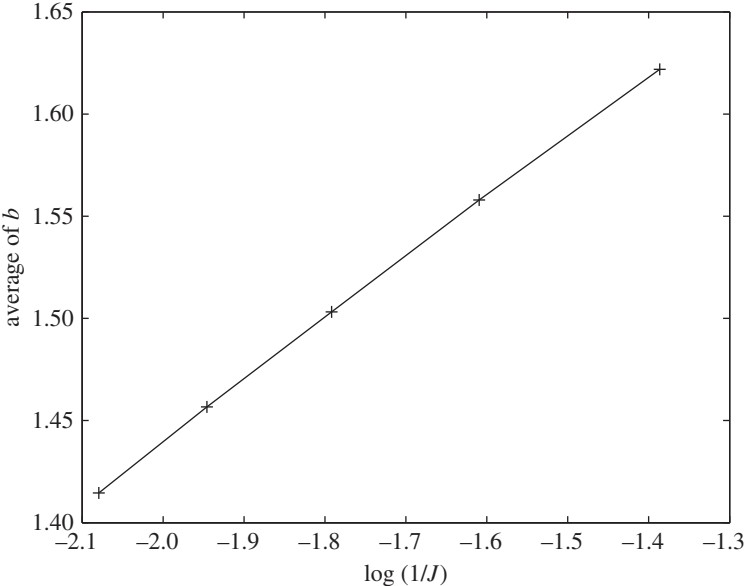

**Figure 4.** The average values of exponent $b$ as a function of the quadrat size. The quadrat size is proportional to $1/J$. The average is taken for different combinations of facilities and cities.

exponent of Taylor's Law at different scales of the quadrats. From figure 2, the exponents are not equal for $J = 5$ and $J = 8$. Case 1 is not true.

Case 2, $b'_i - b'_j = (A/2)(\log (M_i) - \log (M_j))$. $A$ is a constant. From equation (4.4), we have

$$\log (V_i(s)) - \log (V_j(s)) = (b_0 + A \log (s))(\log (M_i(s)) - \log (M_j(s))) \qquad (4.5)$$

$$= b(s)(\log (M_i(s)) - \log (M_j(s))), \qquad (4.6)$$

so that $b(s) = b_0 + A \log (s)$.

We can test if the above argument is true by using five different values of $J$, i.e. $J = 4, 5, 6, 7, 8$. Since the size of the sub-area is fixed as $10\,000 \times 10\,000$ m, the quadrat for a given $J$ is of size $10\,000/J$. In figure 4, the average value of the exponents of 23 cities (in which we can estimate $b$s for all facilities) are plotted against $\log (1/J)$. The five points, corresponding to the five different choices of $J$, are on a straight line with an upward slope. So, our conjecture is verified, i.e. $b(s) = b_0 + A \log (s)$ with $A > 0$ for urban facilities.

# 5. Decomposition of $b$

As we have mentioned in §3, the differences in socioeconomic conditions of the cities and the distinct features of various facilities may result in the fluctuation of the exponent $b$. In order to explain the fluctuation in the values of $b$ for various facilities and in different cities, we can decompose the contributor affecting the number of a facility in the quadrats of a city into two major contributors: a city-specific contributor and a facility-specific contributor. Through the ordinary least-square regression, this decomposition may remove a proportion of estimation error due to insufficient sample quantity.

Assume: (a) the number of a facility in a quadrat of a city is jointly determined by the city-specific contributor and facility-specific contributor, which are assumed to be independent from each other; (b) these two contributors satisfy the following equation for decomposition: $X_{ij} = Y_i Z_j$, where: (a) $X_{ij}$ stands for the quantity of the facility $j$ ($j = 1, 2, \ldots, 7$), in a quadrat of city $i$ ($i = 1, 2, \ldots, 23$); (b) $Y_i$ represents the contribution from city-specific contributor and (c) $Z_j$ represents the contribution of facility-specific contributor. Here we do not specify any measurable variable or quantity for city-specific contributor or facility-specific contributor. Underlying $Y_i$ or $Z_j$ could be a complex function of many variables. We are only interested in the decomposition of the exponents $b$ instead of the specific function form of the factors. Note that the independence and multiplicative

assumption is supported by the results in table 7 that large cities tend to have more facilities of all types than small cities.[11]

From the independent assumption, it is clear that the mean value of $X_{ij}$ could be expressed as follows:

$$M_{ij} = E(X_{ij}) = E(Y_i)E(Z_j) = M_{Y_i}M_{Z_j}, \tag{5.1}$$

where $M_{Y_i} = E(Y_i)$ and $M_{Z_j} = E(Z_j)$, which represents the average contribution of the city-specific contributor and that of the facility-specific contributor respectively. We further assume that there is a power function relationship with exponent $c_i$ between the variance $V$ and the average contribution of the city-specific contributor, while the power function for facility-specific contributor is with exponent $f_j$. This assumption can be represented by the following two equations:

$$M_{Y_i} = \frac{(V_{ij})^{c_i}}{a_{Y_i}} \tag{5.2}$$

and

$$M_{Z_j} = \frac{(V_{ij})^{f_j}}{a_{Z_j}}. \tag{5.3}$$

Based on Taylor's power function $V = aM^b$, we can derive

$$M_{ij} = \frac{(V_{ij})^{1/b_{ij}}}{a^{1/b_{ij}}}. \tag{5.4}$$

Hence, based on equations (5.1)–(5.4), we can infer for the variance part

$$(V_{ij})^{1/b_{ij}} = (V_{ij})^{c_i+f_j}, \tag{5.5}$$

Thus,

$$\frac{1}{b_{ij}} = c_i + f_j + \varepsilon_{ij}, \tag{5.6}$$

where $\varepsilon_{ij}$ represents the estimation error in $b_{ij}$. We assume that $\varepsilon_{ij}$ is normally distributed and has the same variance, i.e. $\varepsilon_{ij} \sim \mathcal{N}(0, \sigma^2)$. The inverse of $b_{ij}$ is decomposed into two components, namely, $c_i$ and $f_j$ in equation (5.6). We call $c_i$ the CSF and $f_j$ the FSF.

It should be noted that we can only solve $c_i$ and $f_j$ up to a constant since they always appear in a summation pair. For example, $c_i - c_0$ and $f_j + c_0$ are also the solutions. It suffices our purpose to examine whether CSFs (or FSFs) take distinct values for different cities (or facilities). To solve equation (5.6), we can define the objective function to minimize:

$$J = \sum_{i=1}^{n_i} \sum_{j=1}^{n_j} \left( \frac{1}{b_{ij}} - (c_i + f_j) \right)^2, \tag{5.7}$$

where $n_i$ and $n_j$ represent the number of cities and that of a facility, respectively. Through minimizing the above objective function, we can derive the value of $c_i$ and that of $f_j$. In appendix B, we give the details on how to solve the problem.

In tables 3 and 4, we list the values of the two factors for all the seven types of facilities in 23 cities under the constraint that $\sum_{i=1}^{n_i} c_i = \sum_{j=1}^{n_j} f_j$ from the Moore–Penrose pseudo inverse. Here, $J = 5$ is used. We notice that $E[|1/b_{ij} - (c_i + f_j)|/(1/b_{ij})] = 7.5\%$, which means that $c_i$ and $f_j$ jointly account for 92.5% of $1/b_{ij}$.

As we can see from table 3, the values of the CSF vary significantly for different cities. The meaning of CSF can be understood in two ways. First, from equation (5.2), these values directly determine the mean value for all facilities within a city, hence they can be seen as indicators of the overall density of all facilities in a given city. Secondly, CSF is a component of $1/b$. Larger CSF means smaller $b$. Smaller $b$ means that the facilities are more evenly distributed. The CSF for Shenzhen is 0.26, which has the highest density of facilities among all cities. It is followed by Beijing (0.24), Shanghai (0.23) and Guangzhou (0.23). These four are the largest cities ranking in the first tier in the Chinese cities. Our results suggest that in larger and more developed cities, we do not only have more facilities available to the citizen, but also the facilities are more evenly distributed so that citizens can better use them.

---

[11]There are some violations in the data, our assumption is an approximation which simplifies our analysis.

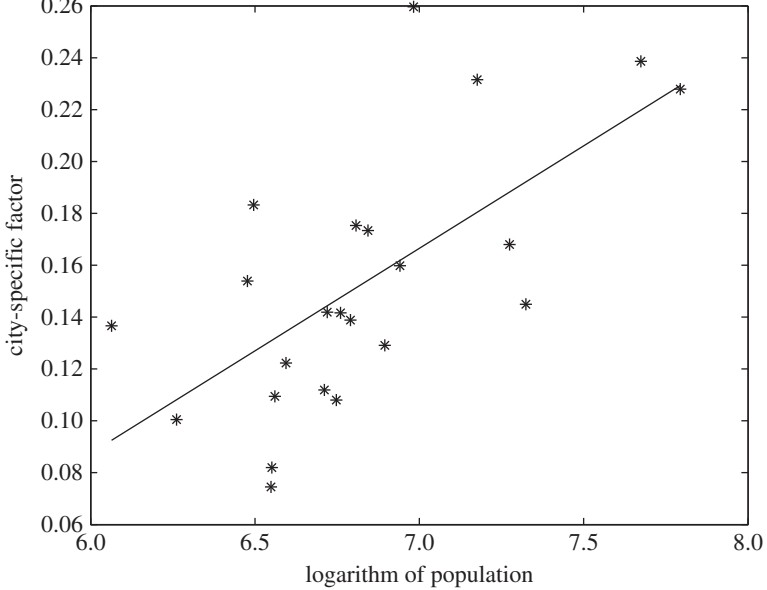

**Figure 5.** Scatter plot of CSF and the logarithm of population of cities.

**Table 3.** The CSF in 23 cities.

| city | Shanghai | Beijing | Tianjin | Chengdu | Guangzhou | Shenzhen |
|---|---|---|---|---|---|---|
| $c_i + c_0$ | 0.23 | 0.24 | 0.15 | 0.17 | 0.23 | 0.26 |
| city | Wuhan | Haerbin | Zhengzhou | Qingdao | Hangzhou | Xian |
| $c_i + c_0$ | 0.16 | 0.13 | 0.17 | 0.18 | 0.14 | 0.14 |
| city | Chongqing | Shenyang | Nanjing | Changsha | Jinan | Xiamen |
| $c_i + c_0$ | 0.11 | 0.14 | 0.11 | 0.12 | 0.11 | 0.08 |
| city | Dalian | Kunming | Wuxi | Nanchang | Taiyuan | |
| $c_i + c_0$ | 0.07 | 0.18 | 0.15 | 0.10 | 0.14 | |

**Table 4.** The FSF for the seven types of facilities.

| urban facilities | beauty salons | banks | stadiums | schools | pharmacies | convenience stores | restaurants |
|---|---|---|---|---|---|---|---|
| $f_j - c_0$ | 0.48 | 0.46 | 0.52 | 0.51 | 0.60 | 0.50 | 0.45 |

For the values of the FSF, we need to understand from another perspective. Because $1/b_{ij} = c_i + f_j$, given a value of $c_i$ in a city, the smaller the value of $f_j$, the larger the values of $b_{ij}$ and therefore larger $V_{ij}$ for a given mean $M_{ij}$ in the corresponding city. Larger variances imply greater differences among the numbers of facility $j$ in different quadrats. At some place, the number is small, but at another place, the number can be very large which means that the facility tends to aggregate in space. On the contrary, when $f_j$ becomes larger, given the same value of mean, the variance falls, thus the distribution tends to behave more like the Poisson distribution, which implies a weaker aggregation. In our decomposition, the value of the FSF for restaurants is 0.45, which is the smallest, and for pharmacies it is 0.60, which is the largest. It shows that restaurants have the highest degree of aggregation; while pharmacies have the lowest degree of aggregation. Restaurants with different styles can coexist at one place; however, due to their interchangeability, pharmacies tend to avoid staying close to each other.

**Table 5.** Correlation test between CSF ($c_i$) and logarithm of population.

| method | correlation coefficient | statistics | $p$-value |
|---|---|---|---|
| Pearson's correlation | 0.65 | 3.95 | 0.00 |
| Spearman's correlation | 0.64 | 726 | 0.00 |

**Table 6.** The ANOVA source table: Tests of the CSF and FSF.

| source | sum of square | d.f. | mean square | F | $p$-value |
|---|---|---|---|---|---|
| CSF | $SS_C = 0.38$ | 22 | $MS_C = 0.017$ | 2.12 | 0.044 |
| FSF | $SS_F = 0.34$ | 6 | $MS_F = 0.056$ | 6.88 | 0.00 |
| error | $SS_E = 1.08$ | 132 | $MS_E = 0.0082$ | | |
| total | $SS_{Total} = 1.80$ | 160 | | | |

In figure 5, we present the scatter plot of the CSF versus the logarithm of population size of cities. Overall, the CSF increases with the population size. Table 5 gives the statistical test. The null hypothesis is that CSF has nothing to do with the population, which is rejected at 1% confidence level. Positive correlation coefficients indicate that CSF is positively correlated with the logarithm of population. Since CSF is a component of the inverse of $b$, the positive correlation between CSF and population indicates that $b$ decreases with population. It is worth mentioning that the conclusion is applicable to all type of facilities, since CSF is a component of $b$ for all type of facilities. It should be noted that China is experiencing a rapid urbanization. The population data, reflecting the official record of permanent residents, does not include the migrating population. The latter becomes more and more important as a lot of Chinese people are migrating from rural areas to cities in recent decades. With larger CSFs and smaller $b$s, larger and more developed cities tend to have more facilities (more service) than small cities.

# 6. Statistic test

## 6.1. ANOVA test

As indicated in tables 3 and 4, cities differ in the CSF and facilities differ in the FSF, thus leading to the fluctuation of exponent $b$ for various facilities in different cities.

To further investigate whether these differences have statistical significance, we use the two-way ANOVA test. The two-way ANOVA is an appropriate analysis method for a study with a quantitative outcome and two categorical explanatory variables.

The two-way ANOVA test has the following assumptions: the sample in each cell (i.e. for each combination of levels of the two factors) is independent of the samples in the other cells, the sample in each cell comes from an (approximately) normal distribution and the populations corresponding to each cell have the same variance (from the homogeneous variance assumption). Our structural model for the two-way ANOVA without interaction is the no-interaction (additive) model. The additive model assumes that the effects on the outcome of a particular level change for one explanatory variable does not depend on the level of the other explanatory variable.

The purpose of the ANOVA test is to investigate the dependence of $b$ on two explanatory variables (city and facility). Similar to the decomposition, the statistical test on $b$ for various facilities in different cities is carried out on the inverse of $b$, i.e. $x_{i,j} = 1/b_{i,j}$, similar to the factor decomposition. The test can be called a $23 \times 7$ ANOVA because those are the levels of the two categorical explanatory variables. The ANOVA test has the two null hypotheses as follows:

$H_1$: There is no difference of the values of $b$ in different cities.
$H_2$: There is no difference of the values of $b$ for various facilities.

We report the result in table 6. From table 6, we obtain the following statistics: $F_{22,132} = 2.12$ and $F_{6,132} = 6.88$, which all correspond to $p < 0.05$. We can reject the null hypotheses (there is no difference of the values of $b$ in different cities and there is no difference of the values of $b$ for various facilities) and conclude that both CSF and FSF have significant effects on the exponent $b$ in the power function.

# 7. Discussion and conclusion

Based on the dataset of spatial coordinates of the seven types of facilities in 37 major cities in China, we explore the micro-structure of these cities and study the characteristics of the distribution of urban facilities. We find that there is a power law function relationship $V = aM^b$ between the variance $V$ and mean $M$ of number of facilities in a quadrat. The distribution of urban facilities complies with Taylor's Law. The same facilities in a city may help each other survive, while at the same time, they compete for various resources, which resembles the relationship between the organisms of a species in an area.

Furthermore, in order to study the key factors contributing to the difference between the values of exponent $b$ and explore the mechanism underlying the distribution of urban facilities, we decompose the inverse of exponent $b$ into two different factors contributing to the numbers of facilities in a city, respectively: the CSF and the FSF. We find that the values of the CSF vary significantly between different cities, and different facilities have different degree of agglomeration. It is interesting to note that Beijing, Shanghai, Guangzhou and Shenzhen, the largest and most developed four cities in mainland China, have the largest $b$. Our results suggest that in larger and more developed cities, we do not only have more service facilities available to the citizen, but also the facilities are more evenly distributed so that citizens can better use them. Moreover, restaurants have the highest degree of agglomeration; while pharmacies have the lowest degree of agglomeration. These findings are consistent with our intuitive understandings of these cities and urban facilities.

Economic activities are often geographically concentrated in particular cities or metropolitan areas, and there are many theories explaining why the concentration may occur [24–26]. Ellison & Glaeser [26] assess the importance of natural advantage to geographical concentration, and find that one-quarter of industrial concentration can be explained by observable sources of natural advantage. Audretsch [27] states that 'knowledge is generated and transmitted more efficiently via local proximity, economic activity based on new knowledge has a high propensity to cluster within a geographic region' (p. 18).

The analytical results in this paper are in line with the findings in the literature mentioned in the above paragraph. Beijing, Shanghai, Shenzhen and Guangzhou are generally acknowledged as the four most urbanized cities in China, and our analytical results show that these cities have the highest level of concentration of urban facilities. Glaeser & Kohlhase [22] show evidence that services tend to be located in dense areas because they are more dependent on proximity to costumers than manufacturing industries. Moreover, there is a strong tendency of service industries to locate near their suppliers and customers, because the costs of delivering services are much higher than the costs of delivering goods. City streets enable service providers to readily link with large numbers of their diverse customers, hence they are a good setting for services. Waldfogel [28] reveals that there is a strong pattern of retail establishment sectors, such as restaurants and media, to locate near demographic groups that regularly buy from that sector.

As we have shown in the above analyses, the distribution of urban facilities resembles that of the organisms in ecosystems. Organisms feed on various resources, while facilities 'feed on' consumer demands. Organisms are prone to form groups, but the size of group varies between different species. For example, zebras and wildebeest form large herds, while the lions usually live in small groups. Urban facilities tend to agglomerate in an area, while as we can see from table 3, the degree of agglomeration varies between different facilities. For instance, the value of the FSF for restaurants is 0.45, which is the smallest, and that for pharmacies is 0.60, which is the largest. This shows restaurants have the highest degree of agglomeration; while pharmacies have the lowest degree of agglomeration (or highest degree of dispersion).

It is important for us to carry out further studies on the distribution of urban facilities, and the potential directions could lie in the following three aspects. Firstly, through combining spatial statistics, economic theories and other relevant fields, we could further explore the rationals and mechanisms underlying the distribution of urban facilities, and examine its impact on socioeconomic development in a city and the adjacent regions. Secondly, when we have sufficient panel data, we

could examine the evolution of the distribution of urban facilities over both the time and space, and explore the relationship between the evolution process and the changes in socioeconomic development indicators, such as income *per capita*, population density and health indicator, etc. Lastly, we could explore relevant theoretical frameworks that could help improve the distribution of urban facilities, thus facilitating sustainable development of cities.

Ethics. No special permit was required. The data were collected using an API from Baidu map service.

Data accessibility. The data and Matlab source code have been uploaded to Dryad Digital Repository: https://doi.org/10.5061/dryad.g9b301f [29].

Authors' contributions. L.W. built the model for the size effect of quadrats and the exponent decomposition; he also drafted the manuscript. C.G. analysed the data. X.Y. checked on the data analysis, and also made corrections on the final version of the paper. All authors gave final approval for publication.

Competing interests. We declare we have no competing interests.

Funding. This research was partially supported by the National Social Science Foundation under grant no. 18BJL075, and also by the Sichuan University Innovation fund with grant no. 2018hhs-49. The authors also want to acknowledge the support from the Sichuan Key Social Science Research Foundation, Sichuan County Economic Development Research Center under grant no. xy2018023.

Acknowledgements. We thank anonymous reviewers and editors for their careful reading and constructive suggestions.

# Appendix A. Descriptive statistics of the facilities in each city

In table 7, we report the descriptive statistics, i.e. population and number of facilities for each city.

# Appendix B. Solution of factor decomposition

From the main text, we have obtained the following quadratic object function $J$ to minimize in order to get the CSF $c_i$ and FSF $f_j$,

$$J = \sum_{i=1}^{n_i} \sum_{j=1}^{n_j} \left( \frac{1}{b_{ij}} - (c_i + f_j) \right)^2. \tag{B 1}$$

Take the derivative with respect to each $c_i$ and each $f_j$, we have the following $n_i + n_j$ linear equations:

$$\left. \begin{array}{l} n_j c_i + \sum_{j=1}^{n_j} f_j = \sum_{j=1}^{n_j} \frac{1}{b_{ij}}, \quad i = 1, 2, \ldots, n_i \\[4mm] \sum_{i=1}^{n_i} c_i + n_i f_j = \sum_{i=1}^{n_i} \frac{1}{b_{ij}}, \quad j = 1, 2, \ldots, n_j. \end{array} \right\} \tag{B 2}$$

and

Define a $(n_i + n_j)$ by $(n_i + n_j)$ matrix $\mathbf{M}$ and a $(n_i + n_j)$ by 1 column vector $\mathbf{X}$,

$$\mathbf{M} = \begin{pmatrix} n_j \mathbf{E}_{n_i \times n_i} & \mathbf{1}_{n_i \times n_j} \\ \mathbf{1}_{n_j \times n_i} & n_i \mathbf{E}_{n_j \times n_j} \end{pmatrix} \tag{B 3}$$

and

$$\mathbf{X} = \begin{pmatrix} \mathbf{c} \\ \mathbf{f} \end{pmatrix}, \tag{B 4}$$

where $\mathbf{E}$ is a unit matrix whose diagonal elements are 1 and off-diagonal elements are 0, $\mathbf{1}$ is a matrix whose elements are all 1, and $\mathbf{c}$ and $\mathbf{f}$ is the column vector for CSFs and FSFs, respectively. We further define a $(n_i + n_j)$ by 1 column vector $\mathbf{B}$ with the elements given by

$$\left. \begin{array}{l} \mathbf{B}_i = \sum_{j=1}^{n_j} \frac{1}{b_{ij}}, \quad i = 1, 2, \ldots, n_i \\[4mm] \mathbf{B}_{n_i+j} = \sum_{i=1}^{n_i} \frac{1}{b_{ij}}, \quad j = 1, 2, \ldots, n_j. \end{array} \right\} \tag{B 5}$$

and

With these definitions, equation (B 2) can be written as $\mathbf{MX} = \mathbf{B}$. However, the rank of $\mathbf{M}$ is $n_i \times n_j - 1$, we would have infinite number of solutions for $c_i$ and $f_j$. Indeed, from equation (B 1), we can see that $c_i$ and $f_j$ appear in a summation pair, we can only solve $c_i$ and $f_j$ up to a constant. If $c_i$ and $f_j$ are one possible

**Table 7.** Population and number of facilities in the study region (40 × 40 km around the centre) for each of the major 37 cities in China.

| | beauty salons | banks | stadium | schools | pharmacies | convenience stores | restaurants | population (million) |
|---|---|---|---|---|---|---|---|---|
| Shanghai | 20 049 | 2919 | 2711 | 5298 | 2918 | 13 938 | 45 900 | 24.26 |
| Beijing | 19 161 | 3485 | 3471 | 6352 | 3686 | 9103 | 42 707 | 21.52 |
| Tianjin | 5486 | 1743 | 762 | 2110 | 2127 | 2084 | 5547 | 15.17 |
| Chengdu | 11 869 | 1631 | 815 | 3347 | 4743 | 9521 | 28 994 | 14.43 |
| Guangzhou | 10 584 | 2059 | 1313 | 3664 | 4401 | 7572 | 21 724 | 13.08 |
| Shenzhen | 9159 | 1157 | 1127 | 2009 | 4304 | 8221 | 19 773 | 10.78 |
| Shijiazhuang | 2971 | 783 | 408 | 1805 | 1327 | 2788 | 6977 | 10.62 |
| Wuhan | 6728 | 1398 | 969 | 3056 | 3072 | 4453 | 15 996 | 10.34 |
| Haerbin | 3968 | 935 | 658 | 2021 | 2606 | 4850 | 13 027 | 9.87 |
| Zhengzhou | 5035 | 812 | 771 | 2703 | 2050 | 4630 | 11 661 | 9.38 |
| Wenzhou | 2762 | 737 | 240 | 1466 | 1167 | 1979 | 4866 | 9.07 |
| Qingdao | 3725 | 806 | 600 | 1569 | 1598 | 2913 | 10 896 | 9.05 |
| Hangzhou | 8059 | 1443 | 902 | 2225 | 2555 | 4088 | 17 598 | 8.89 |
| Xian | 5901 | 1279 | 773 | 2426 | 2278 | 5028 | 17 756 | 8.63 |
| Chongqing | 7439 | 1381 | 545 | 2000 | 3718 | 3644 | 21 966 | 8.52 |
| Shenyang | 7119 | 999 | 1116 | 1453 | 3405 | 8170 | 15 652 | 8.29 |
| Nanjing | 6729 | 1122 | 900 | 2192 | 1655 | 2888 | 17 837 | 8.22 |
| Ningbo | 2876 | 919 | 330 | 1177 | 1010 | 1772 | 7033 | 7.81 |
| Hefei | 3324 | 658 | 376 | 1693 | 1130 | 2295 | 8593 | 7.70 |
| Changchun | 4075 | 799 | 712 | NA | 2186 | 3813 | 10 020 | 7.68 |
| Fuzhou | 3319 | 720 | 467 | 1465 | 947 | 2211 | 7661 | 7.43 |
| Changsha | 4286 | 1080 | 636 | 2095 | 2422 | 5795 | 12 779 | 7.31 |
| Jinan | 3203 | 869 | 556 | 1607 | 1365 | 2361 | 10 337 | 7.07 |

**Table 7.** (*Continued.*)

| | beauty salons | banks | stadium | schools | pharmacies | convenience stores | restaurants | population (million) |
|---|---|---|---|---|---|---|---|---|
| Xiamen | 3889 | 530 | 590 | 1238 | 1007 | 1935 | 9621 | 7.00 |
| Dalian | 4813 | 938 | 613 | 1470 | 1654 | 5011 | 9522 | 6.98 |
| Nanning | 4047 | 616 | 414 | 1860 | 1221 | 2937 | 7782 | 6.91 |
| Kunming | 4544 | 954 | 378 | 1777 | 1656 | 4815 | 11 342 | 6.63 |
| Wuxi | 3414 | 817 | 427 | 1025 | 1265 | 2029 | 8976 | 6.50 |
| Nanchang | 2626 | 755 | 409 | 2004 | 795 | 2491 | 7093 | 5.24 |
| Guiyang | 2324 | 678 | 181 | 1331 | 1354 | 2062 | 6709 | 4.56 |
| Taiyuan | 2672 | 757 | 482 | 1516 | 1113 | 3693 | 8684 | 4.30 |
| Xining | 1617 | 475 | 60 | 514 | 549 | 2905 | 6287 | 3.81 |
| Wulumuqi | 3025 | 654 | 248 | 1011 | 1330 | 3688 | 8521 | 3.53 |
| Huhehaote | 2382 | 579 | 258 | 961 | 911 | 1575 | 6829 | 3.03 |
| Yantai | 1501 | 410 | 195 | 646 | 901 | 1458 | 3517 | 2.29 |
| Yinchuan | 2023 | 454 | 211 | 726 | 719 | 2101 | 4983 | 2.13 |
| Lasa | 306 | 79 | 56 | 138 | 138 | 747 | 1651 | 0.56 |

solution, so are $c_i - c_0$ and $f_j + c_0$. We need one more condition to determine the exact value of $\mathbf{X}$. In this paper, we report the solution of $\mathbf{X}$, which has the smallest L2 norm among all solutions

$$\mathbf{X} = \mathrm{pinv}(\mathbf{M})\mathbf{B}, \tag{B 6}$$

where pinv is the Moore–Penrose pseudo inverse of a matrix. The Moore–Penrose pseudo inverse is solved under the constraint to minimize L2 norm of $\mathbf{X}$, which is to minimize $\sum_{i=1}^{n_i} c_i^2 + \sum_{j=1}^{n_j} f_j^2$ by choosing $c_0$. In this case, it can be shown that the L2 norm constraint is equivalent to $\sum_{i=1}^{n_i} c_i = \sum_{j=1}^{n_j} f_j$.

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
