## [Reviewer comments · Royal Society Open Science]

Review History

RSOS-170738.R0 (Original submission)

Review form: Reviewer 1 (Alastair Greig)

Is the manuscript scientifically sound in its present form?

Yes

Are the interpretations and conclusions justified by the results?

Yes

Is the language acceptable?

Yes

Is it clear how to access all supporting data?

Yes

Do you have any ethical concerns with this paper?

No

Have you any concerns about statistical analyses in this paper?

No

Recommendation?

Major revision is needed (please make suggestions in comments)

Comments to the Author(s)

The overall logic behind this paper is sound. Facilities in human settlements aggregate in cities. However, I feel the paper lacks some more thought provoking conclusions.

In this sense, I'd say the paper might fit into an ecological strand within the Royal Society's publications if it can be shown why the results are relevant to ecologists. The question would be why this is relevant to scientists looking at other species. In terms of a social science paper, I'd say more work needs to be done to show the linkages to population aggregation (are facilities in areas where the population are? do cities with higher aggregation of people have higher aggregation of services?)

In this context, I'd encourage the authors to continue down the line of thinking they have opted for, but a focus on relevance to the audience and concentrating on getting meaningful results from their analysis. A lot of the graphs could be excluded, for example, and this could make way for a more thorough examination of the implications of the descriptive statistics or mechanisms that are driving the results.

Review form: Reviewer 2

Is the manuscript scientifically sound in its present form?

No

Are the interpretations and conclusions justified by the results?

No

Is the language acceptable?

Yes

Is it clear how to access all supporting data?

Yes

Do you have any ethical concerns with this paper?

No

Have you any concerns about statistical analyses in this paper?

Yes

Recommendation?

Major revision is needed (please make suggestions in comments)

Comments to the Author(s)

This paper takes an interesting approach of looking at the spatial agglomeration of facilities in cities, using Taylor's law. The paper is certainly of interest, nevertheless, revisions need to be

undertaken, since there are many aspects that are unclear and might need further testing for statistical significance.

The primary thing that needs to be taken care of is the structure. For a paper in a journal of this level, one would expect that the sections are properly labelled. The introduction seems to indicate that the paper is about the generative mechanisms that give rise to Taylor's law, and this is certainly not the case. The introduction is very similar to the Wikipedia entry on Taylor's law. The bibliography is limited, and some of it has its content wrongly cited, e.g. it is in the Bettencourt paper that the cycle of innovation is discussed. In addition, the discussion of the results comes too soon in the introduction.

There are many issues that need clarification.

Part1:

Data:

- 1) How were the 37 cities selected? How representative of the cities in China are they? Which cities are these? One sees them later on in a plot, but it would be important to give a graph or table with the cities ordered by population, giving the urban extent (area) and number of facilities per classification.
- 2) What is the size of the samples per city?
- 3) Why there were 300 samples chosen? I guess this is only for illustrative purposes.
- 4) The definition of the coordinates has a very bad notation.

Methodology:

- 1) The results might be biased given the area selected for study. Depending of the size of the city the concentration of facilities will differ. Therefore I will strongly suggest the following:
 - a. Select the whole urbanised area for all cities, even of this gives rise to different areas. To do this use satellite images or something of the sort.
 - b. Grid the areas accordingly
 Check whether the results are sensitive to the chosen grid:
 - c. Once a grid has been defined: use a moving window to see whether there's an effect of the position of the grid
 - d. Use different sizes of grid, and compare exponents. So this means that you will be changing I and j , in page 5.

- 2) Why would you choose variables that not all cities contain, and reduce the sample from 30 to 21?
- 3) It is unclear whether when computing S and m , for cities it's the big grid of 16 boxes that is used or the other one. The same applies for individual cities.
- 4) There should be a variable that would serve as the control variable, that is a variable that does not agglomerate in space such that $b=1$.

Results:

- 1) Why are there values for $m < 1$ in figure 2?
- 2) Include the exponent in the scatter plots and CI.
- 3) Downing (Nature 1986) was very critical of comparing the value of the exponent given the uncertainty in its computation. This issue has reappeared over and over again in the measure of scaling laws, see Shalizi, Clauset, Sornette, Cristelli, Arcaute, Cottineau, and more recently Leitao et al. And so the first question that comes to my mind is whether the exponent is smaller for bigger cities, which seems to be the case. In your figure 4, you should order the cities according to size. It would also be interesting to see a plot of size vs value of the exponent., and see whether there's a pattern.
- 4) Selection of facilities. In addition to the previous comment with respect to the control variable, there should be an explanation of why such facilities were chosen.

- a. For example, I'm very puzzle to see stadiums. This is certainly a facility that you wouldn't expect to cluster, nevertheless you seem to find that it does, how is that possible? Are there enough to run such an analysis? I wouldn't expect counts > 2 in cities in general, of course there will be exceptions....or maybe it's a misunderstanding of what the variable is
 - b. What about schools? Do you refer to primary schools or what srt of schools? I would expect a homogeneous distribution of schools in the city so that all citizens can have access to them, hence, once again, I wouldn't expect these to cluster. Why do you find they do?
 - c. Same comment for convenient stores.
- 5) The p-value should be inserted in figure 4.

Part 2: decomposition of the factors.

- 1) It is very odd to think that the two factors are independent. If urban agglomerations do take place according to the scaling laws mentioned in the introduction, then in a bigger city (CSF) there will be for example "more than expected" of a certain facility (FSF).
- 2) Moreover, have a look at the paper by Leitao et al in this same journal, where they do propose how the different indicators would vary according to different proposed models, see city model, etc. This certainly is in line with what the authors have in mind.
- 3) There are other papers in which different regimes are also computed for the scaling laws, considering that there's a regime for small, and a regime for big cities.
- 4) How are the factors computed from the data? It is unclear how these are obtained from eq. (8).
- 5) Overall this section is very weak given its lack of clarity, and doesn't look statistically sound.

Conclusions:

Overall there's a lack of discussion of what the implications are for the results found. What does it mean to have a clustering of certain facilities? And which ones should in principle not cluster? What does it mean for the morphology of the city? What does it mean for the citizens? In addition, I noticed that for the bigger cities b is lower and in the conclusions the opposite is stated, maybe I missed something.

Overall, this paper can be a good contribution to the long standing discussion of scaling laws in cities, if it is reviewed and the city size is addressed properly. There's room for interesting discussions to take place, which I'm looking forward to read next time.

Decision letter (RSOS-170738.R0)

25-Aug-2017

Dear Dr wu,

The editors assigned to your paper ("Taylor's Power Law and its decomposition in Urban Facilities") have now received comments from reviewers. We would like you to revise your paper in accordance with the referee and Associate Editor suggestions which can be found below (not including confidential reports to the Editor). Please note this decision does not guarantee eventual acceptance.

Please submit a copy of your revised paper within three weeks (i.e. by the 17-Sep-2017). If we do not hear from you within this time then it will be assumed that the paper has been withdrawn. In exceptional circumstances, extensions may be possible if agreed with the Editorial Office in advance. We do not allow multiple rounds of revision so we urge you to make every effort to fully address all of the comments at this stage. If deemed necessary by the Editors, your

manuscript will be sent back to one or more of the original reviewers for assessment. If the original reviewers are not available we may invite new reviewers.

- Data accessibility

If you wish to submit your supporting data or code to Dryad (<http://datadryad.org/>), or modify your current submission to dryad, please use the following link:
<http://datadryad.org/submit?journalID=RSOS&manu=RSOS-170738>

- Competing interests

- Authors' contributions

- Acknowledgements

- Funding statement

Yours sincerely,
Alice Power
Editorial Coordinator
Royal Society Open Science

on behalf of Miles Padgett
Subject Editor, Royal Society Open Science
openscience@royalsociety.org

Associate Editor's comments:

Comments to the Author:

Reviewers are asking for some reviews including motivation and framing as well as some better explaining of various parts. We hope the reviews are useful for your revision.

Comments to Author:

Reviewers' Comments to Author:

Reviewer: 1

Comments to the Author(s)

The overall logic behind this paper is sound. Facilities in human settlements aggregate in cities. However, I feel the paper lacks some more thought provoking conclusions.

In this sense, I'd say the paper might fit into an ecological strand within the Royal Society's publications if it can be shown why the results are relevant to ecologists. The question would be why this is relevant to scientists looking at other species. In terms of a social science paper, I'd say more work needs to be done to show the linkages to population aggregation (are facilities in areas where the population are? do cities with higher aggregation of people have higher aggregation of services?)

In this context, I'd encourage the authors to continue down the line of thinking they have opted for, but a focus on relevance to the audience and concentrating on getting meaningful results from their analysis. A lot of the graphs could be excluded, for example, and this could make way for a more thorough examination of the implications of the descriptive statistics or mechanisms that are driving the results.

Reviewer: 2

Comments to the Author(s)

This paper takes an interesting approach of looking at the spatial agglomeration of facilities in cities, using Taylor's law. The paper is certainly of interest, nevertheless, revisions need to be undertaken, since there are many aspects that are unclear and might need further testing for statistical significance.

The primary thing that needs to be taken care of is the structure. For a paper in a journal of this level, one would expect that the sections are properly labelled. The introduction seems to indicate that the paper is about the generative mechanisms that give rise to Taylor's law, and this is certainly not the case. The introduction is very similar to the Wikipedia entry on Taylor's law. The bibliography is limited, and some of it has its content wrongly cited, e.g. it is in the Bettencourt paper that the cycle of innovation is discussed. In addition, the discussion of the results comes too soon in the introduction.

There are many issues that need clarification.

Part I:

Data:

- 1) How were the 37 cities selected? How representative of the cities in China are they? Which cities are these? One sees them later on in a plot, but it would be important to give a graph or table with the cities ordered by population, giving the urban extent (area) and number of facilities per classification.
- 2) What is the size of the samples per city?
- 3) Why there were 300 samples chosen? I guess this is only for illustrative purposes.
- 4) The definition of the coordinates has a very bad notation.

Methodology:

- 1) The results might be biased given the area selected for study. Depending of the size of the city the concentration of facilities will differ. Therefore I will strongly suggest the following:
 - a. Select the whole urbanised area for all cities, even of this gives rise to different areas. To do this use satellite images or something of the sort.
 - b. Grid the areas accordingly
- Check whether the results are sensitive to the chosen grid:
 - c. Once a grid has been defined: use a moving window to see whether there's an effect of the position of the grid
 - d. Use different sizes of grid, and compare exponents. So this means that you will be changing I and j , in page 5.
- 2) Why would you choose variables that not all cities contain, and reduce the sample from 30 to 21?
- 3) It is unclear whether when computing S and m , for cities it's the big grid of 16 boxes that is used or the other one. The same applies for individual cities.
- 4) There should be a variable that would serve as the control variable, that is a variable that does not agglomerate in space such that $b=1$.

Results:

- 1) Why are there values for $m < 1$ in figure 2?
- 2) Include the exponent in the scatter plots and CI.
- 3) Downing (Nature 1986) was very critical of comparing the value of the exponent given the uncertainty in its computation. This issue has reappeared over and over again in the measure of scaling laws, see Shalizi, Clauset, Sornette, Cristelli, Arcaute, Cottineau, and more recently Leitao et al. And so the first question that comes to my mind is whether the exponent is smaller for bigger cities, which seems to be the case. In your figure 4, you should order the cities according to size. It would also be interesting to see a plot of size vs value of the exponent., and see whether there's a pattern.
- 4) Selection of facilities. In addition to the previous comment with respect to the control variable, there should be an explanation of why such facilities were chosen.
 - a. For example, I'm very puzzle to see stadiums. This is certainly a facility that you wouldn't expect to cluster, nevertheless you seem to find that it does, how is that possible? Are there enough to run such an analysis? I wouldn't expect counts > 2 in cities in general, of course there will be exceptions....or maybe it's a misunderstanding of what the variable is
 - b. What about schools? Do you refer to primary schools or what sort of schools? I would expect a homogeneous distribution of schools in the city so that all citizens can have access to them, hence, once again, I wouldn't expect these to cluster. Why do you find they do?
 - c. Same comment for convenient stores.
- 5) The p-value should be inserted in figure 4.

Part 2: decomposition of the factors.

- 1) It is very odd to think that the two factors are independent. If urban agglomerations do take place according to the scaling laws mentioned in the introduction, then in a bigger city (CSF) there will be for example "more than expected" of a certain facility (FSF).
- 2) Moreover, have a look at the paper by Leitao et al in this same journal, where they do propose how the different indicators would vary according to different proposed models, see city model, etc. This certainly is in line with what the authors have in mind.
- 3) There are other papers in which different regimes are also computed for the scaling laws, considering that there's a regime for small, and a regime for big cities.
- 4) How are the factors computed from the data? It is unclear how these are obtained from eq. (8).
- 5) Overall this section is very weak given its lack of clarity, and doesn't look statistically sound.

Conclusions:

Overall there's a lack of discussion of what the implications are for the results found. What does it mean to have a clustering of certain facilities? And which ones should in principle not cluster? What does it mean for the morphology of the city? What does it mean for the citizens? In addition, I noticed that for the bigger cities b is lower and in the conclusions the opposite is stated, maybe I missed something.

Overall, this paper can be a good contribution to the long standing discussion of scaling laws in cities, if it is reviewed and the city size is addressed properly. There's room for interesting discussions to take place, which I'm looking forward to read next time.

Author's Response to Decision Letter for (RSOS-170738.R0)

See Appendices A & B.

RSOS-170738.R1 (Revision)

Review form: Reviewer 2

Is the manuscript scientifically sound in its present form?

No

Are the interpretations and conclusions justified by the results?

Yes

Is the language acceptable?

Yes

Is it clear how to access all supporting data?

Yes

Do you have any ethical concerns with this paper?

No

Have you any concerns about statistical analyses in this paper?

Yes

Recommendation?

Major revision is needed (please make suggestions in comments)

Comments to the Author(s)

The paper has been extensively reviewed given the previous comments, and it has improved considerably. Nevertheless, there are points that are still obscured and need addressing. The most pressing point corresponds to the sensitivity of the exponent with respect to the choice of unit of area. This somehow puzzles me, and the authors did the effort to explain the modified exponent according to scaling arguments. Nevertheless, if the density was considered instead, wouldn't one expect the result to be independent of the unit considered for the measurement? Or under which circumstances can we expect the result to be robust and independent of the choice of unit?

On the other hand, there is a bit of inconsistency in the mathematical notation, which makes it difficult to follow the argument. Appendix B needs reviewing, and appendix A is empty.

For less important matters, here's a list:

- 1) The last sentence in the abstract is unclear.
- 2) Which J is used in page 9.
- 3) Define terms in page 10-11.
- 4) Inconsistent notation X_{ij} , from page 5 and page 13.

I am very much in favour for the publication of this paper, I do hope this will not be too lengthy to address.

Review form: Reviewer 3

Is the manuscript scientifically sound in its present form?

No

Are the interpretations and conclusions justified by the results?

No

Is the language acceptable?

Yes

Is it clear how to access all supporting data?

Yes

Do you have any ethical concerns with this paper?

No

Have you any concerns about statistical analyses in this paper?

Yes

Recommendation?

Reject

Comments to the Author(s)

Thank you for the opportunity to review this manuscript. I found the demonstration of Taylor's law from the various building types quite interesting. The authors would be interested to know that Taylor's law was demonstrated more than 2 decades before this instance from the geographic distribution of buildings over a region of human habitation.

From Figures 2 and 4 it seems that there are some spurious effects associated with the Taylor plots and the assessments of the power law exponent b . Despite the use by the Author's of ANOVA and other statistical methods though, I remain unconvinced about the proposed city-specific and facility specific factors. It appears, to me that the data, due to the considerable amount of noise associated with them, have been somewhat over-interpreted. The conclusions that have been drawn do not appear well-justified.

Decision letter (RSOS-170738.R1)

22-Jan-2018

Dear Dr wu:

Manuscript ID RSOS-170738.R1 entitled "Taylor's Power Law and its decomposition in Urban Facilities" which you submitted to Royal Society Open Science, has been reviewed. The comments from reviewer(s) are included at the bottom of this letter.

In view of the criticisms of the reviewer(s), I must decline the manuscript for publication in Royal Society Open Science at this time. However, a new manuscript may be submitted which takes into consideration these comments.

Please note that resubmitting your manuscript does not guarantee eventual acceptance, and that your resubmission will be subject to re-review by the reviewer(s) before a decision is rendered.

You will be unable to make your revisions on the originally submitted version of your manuscript. Instead, revise your manuscript using a word processing program and save it on your computer.

You may also click the below link to start the resubmission process (or continue the process if you have already started your resubmission) for your manuscript. If you use the below link you will not be required to login to ScholarOne Manuscripts.

*** PLEASE NOTE: This is a two-step process. After clicking on the link, you will be directed to a webpage to confirm. ***

https://mc.manuscriptcentral.com/rsos?URL_MASK=e0d42b96e40b4600b198f541ce2d8058

Because we are trying to facilitate timely publication of manuscripts submitted to Royal Society Open Science, your resubmitted manuscript should be submitted by 22-Jul-2018. If you are unable to submit by this date please contact the Editorial Office for options.

Please note that Royal Society Open Science will introduce article processing charges for all new submissions received from 1 January 2018. Charges will also apply to papers transferred to Royal Society Open Science from other Royal Society Publishing journals, as well as papers submitted as part of our collaboration with the Royal Society of Chemistry (<http://rsos.royalsocietypublishing.org/chemistry>). If your manuscript is submitted and accepted for publication after 1 Jan 2018, you will be asked to pay the article processing charge, unless you request a waiver and this is approved by Royal Society Publishing. You can find out more about the charges at <http://rsos.royalsocietypublishing.org/page/charges>. Should you have any queries, please contact openscience@royalsociety.org.

I look forward to a resubmission.

on behalf of Dr Cecilia Mascolo (Associate Editor) and Miles Padgett (Subject Editor)
openscience@royalsociety.org

Associate Editor Comments to Author (Dr Cecilia Mascolo):

It seems that major doubts about the data and the conclusions drawn remain. We cannot accept this manuscript at this stage but you are certainly allowed to resubmit a revised manuscript and go through a completely new review cycle.

Reviewer comments to Author:

Reviewer: 2

Comments to the Author(s)

The paper has been extensively reviewed given the previous comments, and it has improved considerably. Nevertheless, there are points that are still obscured and need addressing. The most pressing point corresponds to the sensitivity of the exponent with respect to the choice of unit of area. This somehow puzzles me, and the authors did the effort to explain the modified exponent according to scaling arguments. Nevertheless, if the density was considered instead, wouldn't one expect the result to be independent of the unit considered for the measurement? Or under which circumstances can we expect the result to be robust and independent of the choice of unit?

On the other hand, there is a bit of inconsistency in the mathematical notation, which makes it difficult to follow the argument. Appendix B needs reviewing, and appendix A is empty.

For less important matters, here's a list:

- 1) The last sentence in the abstract is unclear.
- 2) Which J is used in page 9.
- 3) Define terms in page 10-11.
- 4) Inconsistent notation X_{ij} , from page 5 and page 13.

I am very much in favour for the publication of this paper, I do hope this will not be too lengthy to address.

Reviewer: 3

Comments to the Author(s)

Thank you for the opportunity to review this manuscript. I found the demonstration of Taylor's law from the various building types quite interesting. The authors would be interested to know that Taylor's law was demonstrated more than 2 decades before this instance from the geographic distribution of buildings over a region of human habitation.

From Figures 2 and 4 it seems that there are some spurious effects associated with the Taylor plots and the assessments of the power law exponent b . Despite the use by the Author's of ANOVA and other statistical methods though, I remain unconvinced about the proposed city-specific and facility specific factors. It appears, to me that the data, due to the considerable amount of noise associated with them, have been somewhat over-interpreted. The conclusions that have been drawn do not appear well-justified.

Author's Response to Decision Letter for (RSOS-170738.R1)

See Appendix C.

RSOS-180770.R0

Review form: Reviewer 4

Is the manuscript scientifically sound in its present form?

Yes

Are the interpretations and conclusions justified by the results?

Yes

Is the language acceptable?

Yes

Is it clear how to access all supporting data?

Yes

Do you have any ethical concerns with this paper?

No

Have you any concerns about statistical analyses in this paper?

Yes

Recommendation?

Accept with minor revision (please list in comments)

Comments to the Author(s)

See attached file (Appendix D).

Decision letter (RSOS-180770.R0)

29-Jan-2019

Dear Dr wu

On behalf of the Editor, I am pleased to inform you that your Manuscript RSOS-180770 entitled "Taylor's Power Law and its decomposition in Urban Facilities" has been accepted for publication in Royal Society Open Science subject to minor revision in accordance with the referee suggestions. Please find the referees' comments at the end of this email.

The reviewers and Subject Editor have recommended publication, but also suggest some minor revisions to your manuscript. Therefore, I invite you to respond to the comments and revise your manuscript.

- Ethics statement

- Data accessibility

If you wish to submit your supporting data or code to Dryad (<http://datadryad.org/>), or modify your current submission to dryad, please use the following link:
<http://datadryad.org/submit?journalID=RSOS&manu=RSOS-180770>

- Competing interests

- Authors' contributions

- Acknowledgements

- Funding statement

Because the schedule for publication is very tight, it is a condition of publication that you submit the revised version of your manuscript before 07-Feb-2019. Please note that the revision deadline will expire at 00.00am on this date. If you do not think you will be able to meet this date please let me know immediately.

on behalf of Dr Cecilia Mascolo (Associate Editor) and Miles Padgett (Subject Editor)
openscience@royalsociety.org

Associate Editor Comments to Author (Dr Cecilia Mascolo):

Associate Editor

Comments to the Author:

The paper and the description of the improvements made with respect to previous reviewers' comments are satisfactory in both the direction of making the manuscript more readable and better motivated as well as in terms of explaining the steps and modelling used. However there are a few minor issues highlighted by one reviewer which still need taking care of before the manuscript is published.

Reviewer comments to Author:

Reviewer: 4

Comments to the Author(s)

See attached file.

Author's Response to Decision Letter for (RSOS-180770.R0)

See Appendix E.

Decision letter (RSOS-180770.R1)

08-Feb-2019

Dear Dr Wu,

I am pleased to inform you that your manuscript entitled "Taylor's Power Law and its decomposition in Urban Facilities" is now accepted for publication in Royal Society Open Science.

Kind regards,

on behalf of Dr Cecilia Mascolo (Associate Editor) and Professor Miles Padgett (Subject Editor)
openscience@royalsociety.org

Appendix A

Reply to reviewer 1.

Many thanks for giving us very constructive comments and the opportunity to revise the paper. We have benefited a lot from your comments, and made some improvement in the modified version. The following is the answer to each of the question/comment marked in red.

Comments to the Author(s)

Q. The overall logic behind this paper is sound. Facilities in human settlements aggregate in cities. However, I feel the paper lacks some more thought provoking conclusions.

A. Thanks for your suggestions, we have strengthen the conclusion in the modified version. We extend our discussions and make connections to the study of agglomeration in economical activities. In this way, we hope our paper can have a more targeted audience from social-economical field.

Q. In this sense, I'd say the paper might fit into an ecological strand within the Royal Society's publications if it can be shown why the results are relevant to ecologists. The question would be why this is relevant to scientists looking at other species. In terms of a social science paper, I'd say more work needs to be done to show the linkages to population aggregation (are facilities in areas where the population are? do cities with higher aggregation of people have higher aggregation of services?)

A. Thank you for your thoughtful suggestion.

We agree with the review that the previous version needs to be improved. We extend our study to include the population size to make our results meaningful in social-economical sense. We observe that larger cities have larger city-specific factor and smaller exponent of Taylor's law. Our results suggest that in larger and more developed cities, we do not only have more service facilities available to the citizen, but also the facilities are more evenly distributed so that citizens can better use them.

Q. In this context, I'd encourage the authors to continue down the line of thinking they have opted for, but a focus on relevance to the audience and concentrating on getting meaningful results from their analysis. A lot of the graphs could be excluded, for example, and this could make way for a more thorough examination of the implications of the descriptive statistics or mechanisms that are driving the results.

A. This is the first paper revealing Taylor's law. We would like to keep some of the graphs/tables to give audience a better understanding of the data, and descriptive statistics etc. As suggested by the reviewer, we have extended the

paper in the line of social-economical implications. In addition, we spend some efforts to report some of the new results about Taylor's law. For example, we find that the Taylor's law is driven both by the competitive nature of business, and also the social-economical environment of a city. With larger CSFs and less \$\$\$, larger and more developed cities tend to have more facilities, and the facilities are more even distributed than smaller cities. Competitive interchangeable services, with smaller \$\$\$, tend to distribute away from each other.

Appendix B

Reply to reviewer 2.

Many thanks for giving us very constructive comments and the opportunity to revise the paper. We have benefited a lot from your comments, and made some improvement in the modified version.

Comments to the Author(s)

This paper takes an interesting approach of looking at the spatial agglomeration of facilities in cities, using Taylor's law. The paper is certainly of interest, nevertheless, revisions need to be undertaken, since there are many aspects that are unclear and might need further testing for statistical significance.

Thanks for your encouraging comments. We have made some modifications. The following is the answer to each of the question/comment.

Q. The primary thing that needs to be taken care of is the structure. For a paper in a journal of this level, one would expect that the sections are properly labelled. The introduction seems to indicate that the paper is about the generative mechanisms that give rise to Taylor's law, and this is certainly not the case. The introduction is very similar to the Wikipedia entry on Taylor's law. The bibliography is limited, and some of it has its content wrongly cited, e.g. it is in the Bettencourt paper that the cycle of innovation is discussed. In addition, the discussion of the results comes too soon in the introduction.

Thanks for your suggesting to improve the introduction. We are sorry for the citation error.

There are many issues that need clarification.

Part1:

Data:

- 1) How were the 37 cities selected? How representative of the cities in China are they? Which cities are these? One sees them later on in a plot, but it would be important to give a graph or table with the cities ordered by population, giving the urban extent (area) and number of facilities per classification.
- 2) What is the size of the samples per city?
- 3) Why there were 300 samples chosen? I guess this is only for illustrative purposes.
- 4) The definition of the coordinates has a very bad notation.

1) & 2) We have clarified these things in the modified version. Basically, the 37 cities are the largest Chinese cities. We use a table to report the number of facilities and the population size in each city.

3) You are right about the 300 samples, they are for illustrative purpose. We have clarified it in the modified version. Thank you for pointing out for us.

4) The definition of the coordinates has been modified.

Methodology:

1) The results might be biased given the area selected for study. Depending of the size of the city the concentration of facilities will differ. Therefore I will strongly suggest the following:

a. Select the whole urbanised area for all cities, even of this gives rise to different areas. To do this use satellite images or something of the sort.

b. Grid the areas accordingly

Check whether the results are sensitive to the chosen grid:

c. Once a grid has been defined: use a moving window to see whether there's an effect of the position of the grid

d. Use different sizes of grid, and compare exponents. So this means that you will be changing I and j , in page 5.

2) Why would you choose variables that not all cities contain, and reduce the sample from 30 to 21?

3) It is unclear whether when computing S and m , for cities it's the big grid of 16 boxes that is used or the other one. The same applies for individual cities.

4) There should be a variable that would serve as the control variable, that is a variable that does not agglomerate in space such that $b=1$.

1) The aim here is to select many sub-areas with different density of urban facilities. Sure, there may be better ways to do that. In our analysis, we use a large enough square (40,000meters by 40,000 meters) around a central point (which we decide by checking the satellite map) to cover the city. Then we let the data decide where the city boundary should be drawn, with a criteria to exclude those sub-areas which do not have more than 20 samples in it. (In the modified version, we have explained why 20.)

About the moving window and different size of the grid. In the modified version, we have added a section to test the robustness of our results. We fix the size of the sub-area, but choose different J , so that we have different quadrat size. Our results are robust with the moving window. However, thanks to the reviewer, we find that the exponent b increase linear with the logarithm. To our surprise, it is a naturally derived result by assuming that Taylor's law holds at different size of the quadrat.

2) For statistical test and decomposition purpose, we need the cities which we can estimate b for all facilities. However, for some cities, we can not. So we end up reducing the number of city from 37 to 23 in the modified version with a more natural criteria. The criteria is, first from the potential 16 sub-areas, we exclude those which does not contain more than 20 samples so that we can not have a good estimation of the mean and variance. Then we decide that if the total number of valid sub-areas is small, e.g., less than five (i.e., we have less than five pairs of means and variances), we can not get a reasonably good

estimation of the mean-variance relationship.

3) Thanks for pointing the problem for us. We have made a clarification. "For each of the 16 sub-areas, we calculate the mean and variance of the number of a facility, thus getting a maximum of 16 pairs of mean and variance."

4) It is a very good idea to have a control variable. However, we can not get one. The city is a fractal. I am not sure. Maybe there is no such variable with uniform distribution.

Results:

1) Why are there values for $m < 1$ in figure 2?

2) Include the exponent in the scatter plots and CI.

3) Downing (Nature 1986) was very critical of comparing the value of the exponent given the uncertainty in its computation. This issue has reappeared over and over again in the measure of scaling laws, see Shalizi, Clauset, Sornette, Cristelli, Arcaute, Cottineau, and more recently Leitao et al. And so the first question that comes to my mind is whether the exponent is smaller for bigger cities, which seems to be the case. In your figure 4, you should order the cities according to size. It would also be interesting to see a plot of size vs value of the exponent., and see whether there's a pattern.

4) Selection of facilities. In addition to the previous comment with respect to the control variable, there should be an explanation of why such facilities were chosen.

a. For example, I'm very puzzle to see stadiums. This is certainly a facility that you wouldn't expect to cluster, nevertheless you seem to find that it does, how is that possible? Are there enough to run such an analysis? I wouldn't expect counts > 2 in cities in general, of course there will be exceptions....or maybe it's a misunderstanding of what the variable is.

b. What about schools? Do you refer to primary schools or what srt of schools? I would expect a homogeneous distribution of schools in the city so that all citizens can have access to them, hence, once again, I wouldn't expect these to cluster. Why do you find they do?

c. Same comment for convenient stores.

5) The p-value should be inserted in figure 4.

1)Because in some sub-area, the total number of samples is less than the number of quadrats (for $J=5$, we have 25 quadrats). On average, there is less than 1 sample in each quadrat, which gives us $m < 1$.

2)Exponents and facility-specific factors are included in the scatter plot. Thanks for your suggestion.

3)We have organize the figure by population size. Now, we can see from the figure (although very noisy), larger cities tend to have smaller b.

4) We focus our study on services, because they tend to be located in dense areas {glaeser2004}. Besides, we only consider the private sector, due to its evolving competitive nature driven by the market. Therefore, we exclude hospitals since many health facilities are state owned. The considered facilities in this study are representative in the sense that they are related to our daily lives in different aspects.

The exact meaning of the facilities is clarified in Table 1. Stadiums here refer not only large comprehensive sports arena, but also those specialized small sports facilities. Also about schools, we refers to all types of schools, from kindergartens to other educational facilities. Otherwise, we won't have enough samples.

Exponent b and p -value are included in the new version.

Part 2: decomposition of the factors.

1) It is very odd to think that the two factors are independent. If urban agglomerations do take place according to the scaling laws mentioned in the introduction, then in a bigger city (CSF) there will be for example "more than expected" of a certain facility (FSF).

The independence and multiplicative assumption is actually supported by the results in Table appendix B. 7, that large cities tend to have more facilities of all types than small cities. Of course, there are some violations in the data, our assumption is an approximation which simplifies our analysis.

2) Moreover, have a look at the paper by Leitao et al in this same journal, where they do propose how the different indicators would vary according to different proposed models, see city model, etc. This certainly is in line with what the authors have in mind.

3) There are other papers in which different regimes are also computed for the scaling laws, considering that there's a regime for small, and a regime for big cities.

To both 2) and 3),

As suggested by Leitao, city-specific observations scale nonlinearly with population. We extend our study to include the population size to make our results meaningful in social-economical sense. We find that larger and more developed cities tend to have smaller b 's. It means that in larger and more developed cities, we do not only have more service facilities available to the citizen, but also the facilities are more evenly distributed so that citizens can better use them.

4) How are the factors computed from the data? It is unclear how these are obtained from eq. (8).

We have given how to compute the factors in the appendix. Thanks f

5) Overall this section is very weak given its lack of clarity, and doesn't look statistically sound.

We agree with reviewer that there is plenty of room to improve in the previous version. We have made improvement in the modified version, and hope that it looks better now.

Conclusions:

Overall there's a lack of discussion of what the implications are for the results found. What does it mean to have a clustering of certain facilities? And which ones should in principle not cluster? What does it mean for the morphology of the city? What does it mean for the citizens? In addition, I noticed that for the bigger cities b is lower and in the conclusions the opposite is stated, maybe I missed something.

We have spent some efforts to report some of the new results about Taylor's law and the implications. For example, we find that the Taylor's law is driven both by the competitive nature of business, and also the social-economical environment of a city. We find that, with larger CSFs and less b s larger and more developed cities tend to have more facilities, and the facilities are more even distributed than smaller cities. Competitive interchangeable services, with smaller b s, tend to distribute away from each other.

About the meaning of our results to the citizens, we extend our analysis to include population size. Our results suggest that in larger and more developed cities, we do not only have more service facilities available to the citizen, but also the facilities are more evenly distributed so that citizens can better use them.

Overall, this paper can be a good contribution to the long standing discussion of scaling laws in cities, if it is reviewed and the city size is addressed properly. There's room for interesting discussions to take place, which I'm looking forward to read next time.

Appendix C

Thanks for giving us the opportunity to revise and resubmit the paper. We have made some changes according to the reviewer's comments and suggestions.

Reply to the reviewer 2

Q1.

First of all, we find that the exponent b does depend on the size of the quadrat. Particularly, the exponent increases linearly with the logarithm of the size. $b(s) = b_0 + A \log(s)$. We derive this result from the assumption that, at different scale, Taylor's law still holds, plus the other scaling rule as discussed by Wu (2014).

Secondly, The exponent is independent of the choice of unit. What we mean is that for a given size of the quadrat, we can either use meter or centimeter to measure the size, we always have the same exponent. In the formula, $b(s) = b_0 + A \log(s)$. s is dimensionless, which does not have a unit. s is relative, measuring the ratio of the size of a quadrat relative to the base quadrat for which we let $s=1$. For instance, at two different scale, we pick one and let the size to be $s=1$, the size of the other one relative to the first one is s . The formula tells how the exponents at the two different scales are related.

Q2.

Thanks for pointing out the problem for us, we have rewritten the mathematical notation and reorganize the description in Appendix B. The modified version should be easier to follow. We add a paragraph to appendix A to show that the purpose of Appendix A is to report Table 6A.

Q3.

We have addressed other minor issues. The abstract has been rewritten. $J=5$ in Fig. 3. Terms have been carefully defined in section 4. The inconsistent issue with notation X_{ij} has been resolved.

Reply to reviewer 3.

Thanks for your question. Please let us clarify your concern.

For your first concern, Although Taylor's law is not new, we have documented two new results regarding the Taylor's law for urban facilities. The exponent b increases linearly with the logarithm of the quadrat size, i.e., $b(s) = b_0 + A \log(s)$.

Furthermore, we decompose b into two different factors, city-specific factor (CSF) and facility-specific factor (FSF). The two factors capture different characteristics of urban facilities.

For your second concern. We agree it is misleading as we see Fig 2 in our original version. We stack the scatter plot of over 30 cities together, we do not have a clear power law relationship. The reason is that the exponent is different for different cities. But if we look at the scatter plot of the mean and variance for one facility in one city, it is clearly a straight line in log-log plot as in the original Fig 3 (Fig.2 in current version). The original Fig 4, now Fig. 3, may seem to be very noisy. Actually, it is not. After we decompose the exponent into two factors, more than 92% of the difference can be explained by the two factors. The data is actually quite clean. So we believe with large amount of data which is available only in recent years thanks to the digital map, we have found something which is not possible two decades ago.

Appendix D

The authors analyse the distribution of retail stores in 37 Chinese cities. First, they show that the spatial distribution of stores can be well described using Taylor's power-law, an empirical law describing the distribution of organisms in ecology. Then, they show that variations in the exponent of the Taylor's power-law can be explained to a large extent by the differences between cities and types of facilities. Stores are more evenly distributed in larger cities, and the distribution ranges from quite even for certain types of stores (i.e. pharmacies) to very aggregated for others (i.e. restaurants).

The Taylor's law has found application in a broad range of fields and the authors demonstrate its applicability in the study of urban infrastructure. The article is clearly written and results are overall demonstrated rigorously. I judge the article is suitable for publications after the following comments are addressed:

As a general comment:

The motivations driving the study should be more clearly stated in the introduction, and possible mechanisms underlying the observed Taylor's behaviour should be better discussed. The conclusions hint that the Taylor's-like behaviour may arise from the interplay between cooperation and competition for resources, but this explanation can be further developed.

In detail:

Line 22, abstract. "Facilities" could be replaced with "types of facilities".

Line 30, Page 3. Please briefly state what is Baidu.

Figure 1. The figure is not easy to interpret since markers are superposed. Maybe one could show different maps for each type of facilities?

Page 5, Line 8. Please explain what is Baidu and how are facilities data collected by Baidu.

Page 5, Line 10. Seven types of facilities.

Page 5, line 18. Typo, list should be listed.

Formula (1). The coordinates inside the Distance function should be $(\text{lng}_i, \text{lat}_i)$, $(\text{lng}_0, \text{lat}_0)$

Page 5, line 40. Please discuss if and how the choice of 40,000 m square affects results.

Page 5, line 43. "And then let the data decides the city boundary". Please clarify this sentence.

Page 5 line 43. Please discuss if and how changing the choice of dividing into 16 squares affects results. Could it be possible to get more statistics the authors could by considering instead a large number of randomly centered squares or circles?

Page 5, line 30. How does the choice of J changes the results? (If this is explained later in the article, please refer the reader to the corresponding section).

Page 7, Line 56. This is result is presented in the introduction as one of the central findings, hence the increase in b should be tested more rigorously. Also, is this true for all types of facilities?

Page 8. Can you clarify in which cases, $b < 1$?

Page 9, Table 2. Please explain why the exponent b systematically decreases for increasing values of J.

Page 9, Line 31. Although the t-test suggests no difference between b and b_s , b_s is consistently smaller than b. Is there a reason for that?

Page 9, line 43. Please explain how the proposition is derived or refer the reader to the literature.

Page 10, line 6. Typo, quarats.

Page 10. Please rephrase and clarify the first paragraph, which is not clear. What do you mean by two events?

Equations (3) and (4). The notation used in the demonstration is not clear. V_i is a function of s, and its value for $s = 1$ is also called V_i .

Page 10, line 38. Please explain what is the reason why we observe that b_i' is different from b_j' .

Page 10, line 51. Typo, fot.

Page 13, line 52. Table 3?

Page 13, line 52 and 56. Here you suggest that $CSF = c_i$? I am confused about the difference between the two.

Page 14, line 6. Evenly.

I suggest the authors may cite this paper, that suggests similar conclusions on the distribution of facilities:

Jensen, Pablo. "Network-based predictions of retail store commercial categories and optimal locations." *Physical Review E* 74.3 (2006): 035101.

Appendix E

Dear Editors and reviewer,

Thanks for giving us the opportunity to revise the paper. We have made some changes according to the reviewer's comments and suggestions.

Reviewer comments to Author:

Reviewer: 4

Q. As a general comment:

The motivations driving the study should be more clearly stated in the introduction, and possible mechanisms underlying the observed Taylor' s behaviour should be better discussed. The conclusions hint that the Taylor' s-like behaviour may arise from the interplay between cooperation and competition for resources, but this explanation can be further developed.

A. We have added two paragraphs in the beginning of the paper to explain the motivations.

In detail:

Q. Line 22, abstract. "Facilities" could be replaced with "types of facilities" .

A. Corrected. Thanks for pointing out for us.

Q. Line 30, Page 3. Please briefly state what is Baidu.

A. We have provided an explanation for Baidu, which says "Baidu is the largest search engine and digital map service provider like Google in China."

Q. Figure 1. The figure is not easy to interpret since markers are superposed. Maybe one could show different maps for each type of facilities?

A. Figure 1 is used for illustrative purpose so that readers can have an idea how the data looks like. We think it will take too much space if we use a map for each type of facilities.

Q. Page 5, Line 8. Please explain what is Baidu and how are facilities data collected by Baidu.

A. The answer to the first part is the same as the above. For the second part, we explain how to collect the data, by saying that "Baidu provides a programmable interface to use the digital map. We collect the spatial coordinates data by calling the interface for seven types of service facilities, in the city area and adjacent counties of 37 major cities in China."

Q. Page 5, Line 10. Seven types of facilities.

A. Corrected here and for the rest of the paper.

Q. Page 5, line 18. Typo, list should be listed.

A. Corrected. Thanks for pointing out the error for us.

Q. Formula (1). The coordinates inside the Distance function should be $(\text{lng}_i, \text{lat}_i)$, $(\text{lng}_0, \text{lat}_0)$

A. Formula (1) is correct, we are calculating the x and y coordinate separately.

Q. Page 5, line 40. Please discuss if and how the choice of 40,000 m square affects results.

A. In the modified paper, we have added a footnote to discuss the choice of 40,000m. "The choice of $40,000m \times 40,000m$ for the starting area is because it is large enough to cover most populated area of a city. The choice of this number should not be an issue for most cities. If we make it bigger, newly added area will most likely be identified as non-city zone. For some cities, it is possible that we can get a bigger exponent b if we enlarge the area since we might include less populated adjacent towns."

Q. Page 5, line 43. "And then let the data decides the city boundary" . Please clarify this sentence.

A. Because of the irregularity of the city area, a few of the sub-areas could be corresponding with the depopulated zones. We mark a sub-area as not valid if the total number of a given facility is less than 20. This method naturally decides where we should draw a line for the city boundary according to the concentration of facilities.

Q. Page 5 line 43. Please discuss if and how changing the choice of dividing into 16 squares affects results. Could it be possible to get more statistics the authors could by considering instead a large number of randomly centered squares or circles?

A. To answer the first question, we have added the following footnote, "The choice of 16 is to balance the need to get more data points and less estimation error for the mean-variance pair. If the number is too big, say, 25, we may end up with small number of facilities in each sub-area, which leads to large estimation error of the mean and variance. If the number is too small, say, 9, for some irregular cities, we may not have enough data points to draw a line for the mean and variance."

Q. Page 5, line 30. How does the choice of J changes the results? (If this is explained later in the article, please refer the reader to the corresponding section).

A. A footnote is added, "We discuss how the choice of J affect our results in section 4."

Q. Page 7, Line 56. This is result is presented in the introduction as one of the central findings, hence the increase in b should be tested more rigorously. Also, is this true for all types of facilities?

A. We have added Table 5 to test if b decreases with population. The statistical test is conducted on the CSF, we show that CSF is positively correlated with population. Since CSF is a component of the inverse of b , the positive correlation between CSF and population indicates that b decreases with population. It is true for all type of facilities, since CSF is a component for all type of facilities.

Q. Page 8. Can you clarify in which cases, $b < 1$?

A. There are two cases for pharmacy that $b < 1$, it means that pharmacy in those two cities tends to be more uniformly distributed due to its competitive nature.

Q. Page 9, Table 2. Please explain why the exponent b systematically decreases for increasing values of J .

A. Sorry, I do not have an explanation for that. It is a mathematical result derived from two scaling laws. I haven't understood it intuitively yet.

Q. Page 9, Line 31. Although the t-test suggests no difference between b and b_s , b_s is consistently

smaller than b . Is there a reason for that?

A. I believe it happens by chance. There are only 5 data points, it is not that rare to observe this pattern. Basically, we just compare two locations to place the quadrats, it should not make the exponent got from one random location larger/smaller than the other random point.

Q. Page 9, line 43. Please explain how the proposition is derived or refer the reader to the literature.

A. We state in the modified paper that, "It is a result that can be derived from the combination of two scaling law regarding mean-variance relationship, namely, Taylor's law and size-scaling law as studied in \cite{wu2014scaling}."

Q. Page 10, line 6. Typo, quarats.

A. Corrected, thanks.

Q. Page 10. Please rephrase and clarify the first paragraph, which is not clear. What do you mean by two events?

A. It should be "two locations".

Q. Equations (3) and (4). The notation used in the demonstration is not clear. V_i is a function of s , and its value for $s = 1$ is also called V_i .

A. We have clarified the confusion by saying that, "where $V_i(s)$ and $V_j(s)$ is the variance using the new quadrats (with size s) in sub-area i and j , respectively. Note that V_i and V_j are measured in the base quadrats when $s=1$."

Q. Page 10, line 38. Please explain what is the reason why we observe that b_i' is different from b_j' .

A. b' refers to the exponent of the size-scaling law. It may take different values in different sub-areas with different densities, so we use subscript i and j to refer to the difference.

Q. Page 10, line 51. Typo, fot.

A. Corrected, thanks.

Q. Page 13, line 52. Table 3?

A. That is right. Table 3. Thanks.

Q. Page 13, line 52 and 56. Here you suggest that $CSF = c_i$? I am confused about the difference between the two.

A. Right, We call c_i the city-specific factor (CSF) and f_j the facility-specific factor (FSF).

Q. Page 14, line 6. Evenly.

A. Corrected, thanks.

Q. I suggest the authors may cite this paper, that suggests similar conclusions on the distribution of facilities: Jensen, Pablo. "Network-based predictions of retail store commercial categories and optimal locations." *Physical Review E* 74.3 (2006): 035101.

A. Indeed, related and very interesting work. Cited.